

# Climate Projections over the Great Lakes Region: Using Two-way Coupling of a Regional Climate Model with a 3-D Lake Model

**Pengfei Xue**[1,2,6,7,*], **Xinyu Ye**[1,2], **Jeremy S. Pal**[3,4], **Philip Y. Chu**[5], **Miraj B. Kayastha**[1], and **Chenfu Huang**[2]

[1]Department of Civil, Environmental and Geospatial Engineering, Michigan Technological University, Houghton, MI

[2]Great Lakes Research Center, Michigan Technological University, Houghton, MI

[3]Department of Civil Engineering and Environmental Science, Loyola Marymount University, Los Angeles, California

[4]Risk Assessment and Adaptation Strategies Division, Euro-Mediterranean Center on Climate Change and Ca' Foscari University, Venice, Italy

[5]NOAA/Great Lakes Environmental Research Laboratory, Ann Arbor, Michigan

[6]Environmental Science Division, Argonne National Laboratory, Lemont, IL

[7]Department of Civil and Environmental Engineering, Massachusetts Institute of Technology, Cambridge, MA

**Corresponding Author:** Pengfei Xue (pexue@mtu.edu)

**Abstract**
Warming trends of the Laurentian Great Lakes and surrounding areas have been observed in recent
decades, and concerns continue to rise about the pace and pattern of future climate change over the
worlds largest freshwater system. To date, many regional climate models used for the Great Lakes
projection either neglected the lake-atmosphere interactions or only coupled with 1-D column
lake models to represent the lake hydrodynamics. The study presents the Great Lakes climate
change projection that has employed the two-way coupling of a regional climate model with a 3-D
lake model (GLARM) to resolve 3-D hydrodynamics important for large lakes. Using the three
carefully selected CMIP5 AOGCMS, we show that the GLARM ensemble average substantially
reduces the surface air temperature and precipitation biases of the driving AOGCM ensemble
average in present-day climate simulations. The improvements are not only displayed from the
atmospheric perspective but also evidenced in accurate simulations of lake temperature, and ice





coverage and duration. After that, we present the GLARM projected climate change for the mid-21st century (2030-2049) and the late century (2080-2099) for the RCP 4.5 and RCP 8.5. Under RCP 8.5, the Great Lakes basin is projected to warm by 1.3-2.2°C by the mid-21st century and 4.0-4.9°C by the end of the century relative to the early-century (2000-2019). Moderate mitigation (RCP 4.5) reduces the mid-century warming to 0.8-1.9°C and late-century warming to 1.8-2.7°C. Annual precipitation in GLARM is projected to increase for the entire basin, varying from -0.4% to 10.5% during the mid-century and 1.2% to 28.5% during the late-century under different scenarios and simulations. The most significant increases are projected in spring and early summer when current precipitation is highest and little increase in winter when it is lowest. Lake surface temperatures (LSTs) are also projected to increase across the five lakes in all of the simulations, but with strong seasonal and spatial variability. The most significant LST increase will occur in Lake Superior. The strongest warming was projected in spring, followed by strong summer warming, suggesting earlier and more intense stratification in the future. In contrast, a relatively smaller increase in LSTs during fall and winter are projected with heat transfer to the deepwater due to strong mixing and energy required for ice melting. Correspondingly, the highest monthly mean ice cover is projected to be 3-6% and 8-20% across the lakes by the end of the century in RCP 8.5 and RCP 4.5, respectively. In the coastal regions, ice duration will decrease by up to 30-50 days.

**Keywords:** Two-way Coupling; Climate Change; Climate Projection; Great Lakes; Earth System; Model Development

# 1 Introduction

The Laurentian Great Lakes are the world's largest surface freshwater systems, containing 84% of North America's surface freshwater and 21% of the world's supply of surface fresh water (EPA 2014). Spanning more than 244,000 km$^2$, an area roughly equal to the size of the United Kingdom, the vast inland freshwater system provides water for consumption, transportation, power, recreation, and many other uses. The Great Lakes support 1.3 million jobs and $82 billion in wages per year (Rau et al. 2020). More than 34 million people call the Great Lakes basin home, and more than 3500 species of plants and animals inhabit it, including over 170 species of fish (EPA 2014). The Great Lakes commercial, recreational, and tribal fisheries are collectively valued at more than $7 billion annually and support more than 75,000 jobs (`http://www.glfc.org/the-fishery.php`).

In recent decades, Great Lakes and surrounding areas have undergone rapid warming (Austin and Colman 2007; Dobiesz and Lester 2009; Hayhoe et al. 2010; Melillo et al. 2014; Pryor et al. 2014;





Zhong et al. 2016). The annual mean temperature over the Great Lakes basin has increased by 0.9°C between 1901-1960 and 1985-2016, exceeding average changes of 0.7°C for the rest of the contiguous United States (Wuebbles et al. 2019). Consequently, lake surface temperature (LST) in the Great Lakes has increased and ice coverage has decreased. Summer LST has risen faster than the ambient air temperature in Lake Superior (Austin and Colman 2008; McCormick and Fahnenstiel 1999). Ice coverage has reduced by 71% on the Great Lakes as a whole from 1973 through 2010 (Wang et al. 2012).

Measurable changes have also been observed in precipitation patterns, lake levels, wave climate, and water biogeochemistry impacting the ecosystems (Huang et al. 2021b; Jones et al. 2006; Wuebbles et al. 2019). For example, climate change and human activities have influenced algal bloom frequency and intensity (Dalolu et al. 2012; Dobiesz and Lester 2009; Scavia et al. 2014) reduced primary productivity (Poesch et al. 2016), and altered prey fish habitats and population (Collingsworth et al. 2017; Lynch et al. 2016; Sharma et al. 2007). As a result, there has been a growing need to better understand climate change and variability for the Great Lakes and surrounding regions.

Various techniques have been used to project how the Great Lakes regional climate will evolve in the future. The direct use of coupled Atmosphere-Ocean General Circulation Models (AOGCMs) simulation results has shown various problems due to their typical low spatial resolution resulting in inadequacies in representing small-scale processes important in the region (MacKay and Seglenieks 2013). More importantly, many Coupled Model Intercomparison Project Phase 5 (CMIP5) models do not include credible representations of Great Lakes (Briley et al. 2021). Dynamical downscaling using higher-resolution regional climate models (RCMs) has been used to improve on these inadequacies (e.g., Music et al. 2015; Notaro et al. 2015; Xiao et al. 2018; Zhang et al. 2019, 2018, 2020). Statistical downscaling (Byun and Hamlet 2018; Byun et al. 2019) and probabilistic projection using a Bayesian Hierarchical Model (Wang et al. 2017) have also been recently applied to the Great Lakes region.

Regardless of the techniques used, temperatures over the Great Lakes basin are predicted to increase with anthropogenic atmospheric greenhouse gasses (GHGs) (e.g., Byun and Hamlet 2018; Cherkauer and Sinha 2010; Zhang et al. 2020). Projected precipitation changes are less certain, however, several studies project reductions in summer precipitation and increases in winter and spring, as well as an increase in the fraction of precipitation falling as rainfall (Byun and Hamlet 2018; Cherkauer and Sinha 2010; Notaro et al. 2015; Zhang et al. 2019). Similarly, the lakes themselves are projected to continue to rapidly warm, resulting in reduced ice cover and earlier occurrence of seasonal stratification (Gula and Peltier 2012; Notaro et al. 2015; Xiao et al. 2018). These changes can further modify the distribution of lake mixing regimes and shift the timing of lake overturning episodes (Woolway and Merchant 2019), and can have profound implications for lake biogeochemistry, ecosystems, power production, navigation, tourism, and other sectors.





Uncertainties in Great Lakes climate change projections can arise from multiple sources including GHG emission scenarios, internal variability, model deficiencies and lateral forcing conditions. However, land-lake-ice-atmosphere interactions must be taken into account. While significant improvements have been made in modeling these systems, they are typically modeled independently, loosely coupled, or with only a limited set of interactions. Few previous studies have applied a dynamical approach to downscaling AOGCM for climate change projections with results of changes in Great Lakes conditions (Gula and Peltier 2012; Mailhot et al. 2019; Notaro et al. 2015). However, these studies generally treated the Great Lakes as one-dimensional (1D) water columns and ignored three-dimensional (3D) processes in the large lakes (Bennington et al. 2014; Hostetler et al. 1993; Subin et al. 2012). Incorporating 3D hydrodynamic models into RCMs to represent the hydrodynamics of the Great Lakes has been advocated by the Great Lakes modeling community but still in its early stage (Delaney and Milner 2019). Recently, Xue et al. (2017) developed the first two-way coupled RCM and 3D hydrodynamic model system and demonstrated the feasibility and clear benefit of this approach for regional climate simulation. This approach leads to more accurate representations of surface wind regulated sensible and latent heat fluxes that reduce in LST biases (Xue et al. 2015) and improve the simulation of atmospheric conditions such as precipitation and lake-effect snow due to improved representation of LSTs (Shi and Xue 2019). More recently, a similar study using the Climate-Weather Research and Forecasting Model (CWRF) coupled with FVCOM developed for historical simulations (Sun et al. 2020) also demonstrated improved performance when coupling atmosphere and 3-D lake models in a two-way fashion. These two efforts, however, have focused on model development and validation. To date, no studies exist applying such coupled 3-D two-way coupled models to project evolution of the Great Lakes themselves interacting with regional climate changes.

In this study, a RCM two-way coupled with a 3-D hydrodynamic model to fully resolve the lake-ice-atmosphere interactions is utilized to provide more reliable high-resolution projections of climate change for the Great Lakes and surrounding regions. Ensemble projections are conducted for the mid- and late twenty-first century under a "business as usual" Representative Concentration Pathway (RCP) scenario (RCP 8.5) and a mitigation scenario (RCP 4.5). The paper documents the model development, validation, and climate change projections. Emphasis is placed on the climate change over the Great Lakes basin as well as its impacts on and interactions with the changes within the lakes.

# 2 Model and Numerical Experiment Design

## 2.1 GLARM

The Great LakesAtmosphere Regional Model (GLARM) is a two-way lake-iceatmosphere coupled climate model designed for the Great Lakes region (Xue et al. 2017). GLARM consists of the 4th version of the International Centre for Theoretical Physics (ICTP) Regional Climate Model





(RegCM4) to simulate land and atmospheric processes (Giorgi et al. 2012) and the Finite Volume
Community Ocean Model (FVCOM) to simulate the 3-D lake dynamics, thermal dynamics, and
ice dynamics (Chen et al. 2012). The version of RegCM4 applied in this study is a 3-D, hydrostatic,
compressible, primitive equation, $\sigma$-coordinate and has a nearly identical configuration to RegCM3
(Pal et al. 2007). FVCOM is an unstructured-grid, finite-volume, 3-D, primitive equation, hydrodynamic
model with a generalized, terrain-following coordinate system in the vertical and triangular meshes
in the horizontal, and is widely applied to coastal oceans and the Great Lakes (Anderson et al. 2018;
Huang et al. 2021a, 2019; Ibrahim et al. 2020; Xue et al. 2014, 2020, 2015; Ye et al. 2019, 2020).
GLARM has been configured with a large domain and small domain in this study. The large
domain includes the majority of North America (NA) to fully enable model internal variability
and dynamic consistency (Fig. 1, green box, hereafter referred to GLARM-large). The RegCM4
module (land, atmosphere and ocean) has an 18-km horizontal grid spacing and 18 vertical sigma
layers. The FVCOM module (Great Lakes) has a horizontal resolution of unstructured triangular
grids that varies from  1-2 km near the coast to  2-4 km in the offshore region of the lakes. The
model is configured with 40 sigma layers to provide a vertical resolution of < 1 m for nearshore
waters and  2-5 m in most of the offshore regions of the lakes. The smaller domain is identical in
configuration but limited in coverage to the Midwest and Northeast United States and the Ontario
and Quebec Canadian provinces (Fig. 1, red box, hereafter referred to GLARM-small), comparable
in size to other previous Great Lakes RCM configurations (e.g., Bennington et al. 2014; Xiao et al.
2018). This smaller domain, which may be influenced more by driving AOGCMs through lateral
boundary conditions, serves as a computationally efficient alternative to the larger domain for
comparison.



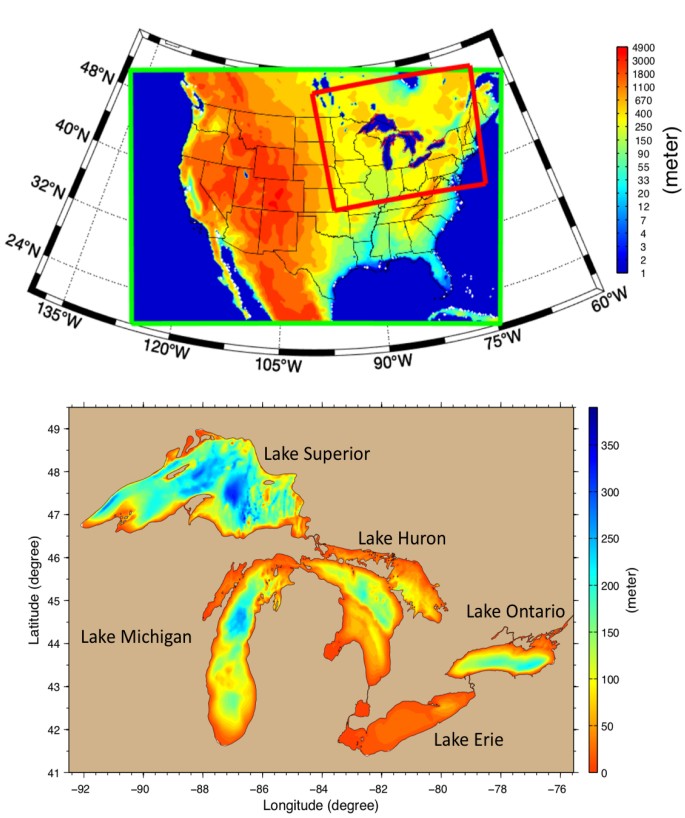

Figure 1: Top: GLARM configured with a large North America domain (green box) and GLARM configured with a smaller domain (red box). Bottom: Bathymetry of the Great Lakes.

## 2.2 Data for Model Validation

Various datasets were used in this study for evaluating the model performance in simulating present-day climate, which is a vital step to produce reliable projections. Monthly surface air temperature and precipitation were obtained from the land-station-based 0.5°Climate Research Unit data (CRU TS 3.0) (Harris et al. 2014) and the daily LSTs for the five lakes from the Great Lakes Surface Environmental Analysis (GLSEA; https://coastwatch.glerl.noaa.gov/glsea/glsea.html). Derived from NOAA/AVHRR (Advanced Very High Resolution Radiometer) satellite imagery, GLSEA serves as the best available product to examine spatial and temporal variability of surface water temperature in the Great Lakes. The daily Great Lakes ice coverage was obtained from the Great Lakes Ice Cover Database (GLICD) using the ice products developed by the U.S. National Ice Center and the Canadian Ice Service (https://www.glerl.noaa.gov/data/





ice/#historical), which includes the Great Lakes Ice Atlas (https://www.glerl.noaa.gov/
data/ice/atlas/) for the period 1973-2002 and ice data addendum for 2003 through present.

## 2.3 Numerical Experiment Design

The Intergovernmental Panel on Climate Change (IPCC) projections are largely based on AOGCM
simulations from the Coupled Model Intercomparison Project (CMIP) coordinated framework.
As configured, the output from these simulations is a credible data source for climate change
assessments at global, continental, and regional scales; however it may not adequately represent
regional and localized features due to the relatively coarse spatial resolution of the AOGCMs (100s
km). Using AOGCMs output to drive RCMs has been shown to enhance model performance due
largely to a more realistic representation of physics and dynamics as well as orography, coastlines,
and land cover as a consequence of their higher resolution. A primary factor of uncertainty
associated with the CMIP5 climate change projections is that different AOGCMs can simulate
very different climate changes across global, continental and regional scales even under the same
anthropogenic forcing scenario. For regional climate modeling studies it is, therefore, critical
to evaluate AOGCM performance in the region of interest and select those that best represent
climate. In this work, we first evaluate the performance of CMIP5 AOGCMs and then select
a subset to use as lateral and ocean surface boundary conditions for GLARM. The GLARM
present-day (2000-2019) simulations, driven by the selected AOGCMs, are then validated against
observational data. As the CMIP5 AOGCM hindcast simulations ended in 2005, the AOGCM
results for 2006-2019 under RCP8.5 were used to drive GLARM for the best track of observed
GHG emission (Schwalm et al. 2020). After that, the GLARM projected climate change for
the mid-21st century (2030-2049) and the end of the century (2080-2099) for the RCP 4.5 and
RCP 8.5 scenarios are presented and discussed. RCP 8.5 is representative of a scenario with
high atmospheric GHG concentrations while RCP 4.5 represents a scenario with considerable
mitigation.
The output from 19 CMIP5 AOGCMs (Table 1) are assessed based on two general reliability
criteria (Giorgi and Mearns 2002). The first criteria is based on the ability of the AOGCMs to
reproduce different aspects of historical climate, referred to as the "model performance" criterion.
The second, referred to as the "model convergence" criterion, assesses the convergence of climate
projections by different models under a given forcing scenario. Higher convergence implies more
robust signals (Giorgi and Mearns 2002). The reliability score $R_k$ represents the $K_t h$ model performance
in simulating the historical climate and its degree of convergence in the projected future climate:

$$R_k = [(R_{B,K})^m \times (R_{D,K})^n]^{\frac{1}{m \times n}} = [(\frac{\varepsilon}{|B_k|})^m \times (\frac{\varepsilon}{|D_k|})^n]^{\frac{1}{m \times n}}, \quad (1)$$






$$\overline{T} = \frac{\sum_{k=1}^{n}(R_K \times T_K)}{\sum_{k=1}^{n} R_K} \qquad (2)$$

$R_{B,k}$ is a factor that is inversely proportional to the absolute bias $B_k$ in simulating the historical
variable and $R_{D,k}$ measures the model convergence in terms of the distance ($D_k$) of the departure
of a given model from the average ensemble change weighted by the reliability score of each
model $R_k$ (i.e., reliability ensemble average or REA). The parameters $m$ and $n$ (typically equal
to 1) represent the weights of the model performance criterion ($R_{B,k}$) and the model convergence
criterion ($R_{D,k}$) that influence the reliability score $R_k$ of the model, respectively. The parameter
$\varepsilon$ describes the natural variability of the climatic variable. $\overline{T}$ is the REA of an assessed variable
(e.g. surface air temperature) based on individual value $T_k$ ($k = 1, 19$). The reliability score $R_k$ is
calculated iteratively to converge, since $R_k$ is a function of REA, and REA in turn is updated with
$R_k$.
To evaluate the performance of each AOGCM in reproducing observed climate and projecting the
future warming trend over NA, the model reliability analysis is conducted using model-simulated
NA-averaged temperature in the historical periods (1901-2005) and the future period (2006-2100)
in RCP 8.5 scenario. The three AOGCMs with the highest reliability scores are selected to drive
GLARM for the present-day and two future periods under each scenario.



Table 1: AOGCMs used for reliability analysis.

| | GCM Model | Institute | Resolution (degree) | |
|---|---|---|---|---|
| | | | Latitude | Longitude |
| 1 | ACCESS1.3 | Commonwealth Scientific and Industrial Research Organization/Bureau of Meteorology, Australia | 1.25 | 1.875 |
| 2 | CNRM-CM5 | Centre National de Recherches Météorologiques, Centre Européen de Recherche et de Formation Avancée en Calcul Scientifique, France | 1.4008 | 1.40625 |
| 3 | GFDL-CM3 | Geophysical Fluid Dynamics Laboratory, NOAA, United States | 2 | 2.5 |
| 4 | GFDL-ESM2G | As above | 2.0225 | 2 |
| 5 | GFDL-ESM2M | As above | 2.0225 | 2.5 |
| 6 | GISS-E2-H | GISS (Goddard Institute for Space Studies), NASA, United States | 2 | 2.5 |
| 7 | GISS-E2-R | As above | 2 | 2.5 |
| 8 | HadGEM2-ES | Met Office Hadley Centre, UK | 1.25 | 1.875 |
| 9 | IPSL-CM5A-LR | Institut Pierre Simon Laplace, France | 1.8947 | 3.75 |
| 10 | IPSL-CM5A-MR | As above | 1.2676 | 2.5 |
| 11 | IPSL-CM5B-LR | As above | 1.8947 | 3.75 |
| 12 | MIROC5 | Atmosphere and Ocean Research Institute, National Institute for Environmental Studies, and Japan Agency for Marine-Earth Science and Technology, Japan | 1.4008 | 1.40625 |
| 13 | MIROC-ESM-CHEM | As above | 2.7906 | 2.8125 |
| 14 | MIROC-ESM | As above | 2.7906 | 2.8125 |
| 15 | MPI-ESM-LR | Max Planck Institute for Meteorology, Germany | 1.8653 | 1.875 |
| 16 | MPI-ESM-MR | As above | 1.8653 | 1.875 |
| 17 | MRI-CGCM3 | Meteorological Research Institute, Japan | 1.12148 | 1.125 |
| 18 | NorESM1-M | Bjerknes Centre for Climate Research, Norwegian Meteorological Institute, Norway | 1.8947 | 2.5 |
| 19 | NorESM1-ME | Bjerknes Centre for Climate Research, Norwegian Meteorological Institute, Norway | 1.8947 | 2.5 |



## 3 Results

### 3.1 AOGCM Evaluation and Selection

Due to the high computational cost of dynamical downscaling progress using the GLARM, downscaling all AOGCMs is not feasible at this time. Therefore a subset of AOGCMs is selected based on the ability of the AOGCM performance in simulating mean surface air temperature over NA. Among the 19 AOGCMs, the IPSL-CM5A-MR, MPI-ECM-MR, and GISS-E2-H received the highest reliability scores (Table 2). To validate the AOGCM selections, we show that our selected three-model ensemble average (AOGCM-EA3) 1) outperformed 19 individual CMIP5 AOGCMs and 2) was comparable to, if not better than, the 19-model ensemble average (AOGCM-EA19) in three performance metrics including correlation coefficient (R), centered root-mean-square deviation (RMSD) and standard deviation (Std) depicted in the Taylor diagram (Fig. 2-a).

These performance metrics are calculated for the 10-year moving average of surface air temperature over NA to evaluate AOGCMs capability of capturing the decadal variation. The scores from the metrics for the 19 AOGCMs span a wide range of values (e.g., R, Std, and RMSD range from 0.45-0.93, 0.15-0.45°C and 0.11-0.33°C, respectively). Both AOGCM-EA19 and AOGCM-EA3 show very similar performance with a smaller RMSD (0.11-0.12°C) and higher correlation (0.90-0.93) than any single AOGCM; thus highlighting the benefit of ensemble climate modeling. In addition, AOGCM-EA3's standard deviation (0.27°C) is closer to the observation (0.28°C) compared to AOGCM-EA19's (0.21°C), thereby providing us with some confidence in the selected three AOGCMs for dynamical downscaling.

In terms of observed warming, the 10-year moving average of annual air temperature for both AOGCM-EA19 and AOGCM-EA3 captures the observed trend, including rapid warming after the 1980s. Additionally, GCM-EA3 tracks the historical temperatures significantly better than GCM-EA19 (Fig. 2-b). The temperatures predicted from GCM-EA3 and GCM-EA19 remain similar to the observations, however after 1930, GCM-EA19 deviates and maintains a nearly constant cold bias of 0.4°C. GCM-EA3, in contrast, closely follows the observation trend and magnitude yielding a mean bias of -0.06°C, which further justifies our selection of the three models.



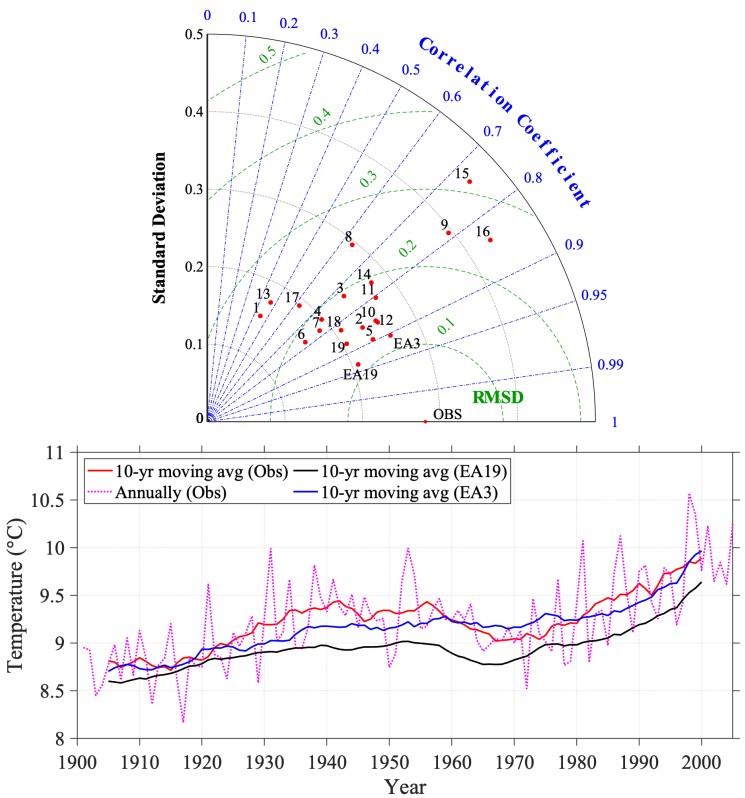

Figure 2: Top: Taylor diagram for 19 individual AOGCMs, ensemble average of 19 AOGCMs (EA19), and ensemble average of the three selected AOGCMs (IPSL-CM5A-MR (10), MPI-ECM-MR (16), and GISS-E2-H (6)) ensemble average (EA3) for the 10-yr moving average of surface air temperature simulation in the period of 1901-2005 over North America. Bottom: Annual surface air temperature (pink), its 10-yr moving average in the period of 1901-2005 comparisons between CRU observations (red), three selected model ensemble average (EA3; blue), and 19-model ensemble average (EA19; black).





Table 2: AOGCMs performance metrics: R, Std, RMSE and model REA score for decadal air temperature simulations over North America in 19 individual AOGCMs and AOGCM-EA19 and AOGCM-EA3.

|  | GCM Model | Correlation (R) | Standard deviation (Std) | RMSD | REA normalized score |
|---|---|---|---|---|---|
| 1 | ACCESS1-3 | 0.44 | 0.15 | 0.25 | 0.044 |
| 2 | CNRM-CM5 | 0.85 | 0.23 | 0.14 | 0.062 |
| 3 | GFDL-CM3 | 0.73 | 0.23 | 0.19 | 0.022 |
| 4 | GFDL-ESM2G | 0.74 | 0.19 | 0.18 | 0.029 |
| 5 | GFDL-ESM2M | 0.89 | 0.23 | 0.12 | 0.042 |
| 6 | GISS-E2-H | 0.77 | 0.16 | 0.18 | 0.113 |
| 7 | GISS-E2-R | 0.77 | 0.18 | 0.17 | 0.059 |
| 8 | HadGEM2-ES | 0.63 | 0.29 | 0.24 | 0.042 |
| 9 | IPSL-CM5A-LR | 0.78 | 0.39 | 0.24 | 0.037 |
| 10 | IPSL-CM5A-MR | 0.85 | 0.25 | 0.14 | 0.119 |
| 11 | IPSL-CM5B-LR | 0.8 | 0.26 | 0.17 | 0.032 |
| 12 | MIROC5 | 0.86 | 0.25 | 0.14 | 0.036 |
| 13 | MIROC-ESM-CHEM | 0.46 | 0.17 | 0.25 | 0.013 |
| 14 | MIROC-ESM | 0.76 | 0.27 | 0.19 | 0.013 |
| 15 | MPI-ESM-LR | 0.73 | 0.45 | 0.31 | 0.097 |
| 16 | MPI-ESM-MR | 0.841 | 0.43 | 0.24 | 0.119 |
| 17 | MRI-CGCM3 | 0.62 | 0.19 | 0.22 | 0.017 |
| 18 | NorESM1-M | 0.82 | 0.2 | 0.16 | 0.056 |
| 19 | NorESM1-ME | 0.87 | 0.2 | 0.14 | 0.05 |
| 20 | GCM-EA19 | 0.93 | 0.2 | 0.11 | — |
| 21 | GCM-EA3 | 0.9 | 0.27 | 0.12 | — |





## 3.2 Dynamical Downscaling using GLARM

Before analyzing the climate change projections, we first verify how well GLARM predicts the present-day (2000-2019) surface air temperature, precipitation, lake surface temperature, and ice cover forced by the selected three AOGCMs (IPSL-CM5A-MR, MPI-ECM-MR, and GISS-E2-H) for both GLARM-large and GLARM-small (3 AOGCMs × 2 domains). The ensemble average of the six-member predictions was hereafter referred to as GLARM-EA6.

### 3.2.1 Present-day Climate

Figure 3 exhibits GLARM's superiority over the selected three GCMs in reproducing the historical air temperature and precipitation over the Great Lakes basin. Both AOGCM-EA3 and GLARM-EA6 reproduce the spatial pattern of observed air temperature well, with the model-data pattern correlations of 0.948 for GLARM-EA6 and 0.987 for AOGCM-EA3 (Fig. 3). However, GLARM-EA6 has a considerably smaller bias (0.18 °C) over the Great Lakes basin compared to AOGCM-EA3 (0.94 °C). The warm bias produced by the AOGCM-EA3 for the northern parts of the basin is notably reduced in GLARM-EA6 (Fig. 3-c1,c2). It should be noted that the CRU data inaccurately represents air temperature over the lakes since it is land station based. As all of the selected AOCMs considered ignore or only provide crude representations of the Great Lakes (Fig. 3-b2), the temperature patterns over land and over lake are quite similar. Unlike the GCM-EA3 simulations, GLARM-EA6 simulations indeed manifest the lake influence on the over-lake air temperatures, reinforcing the importance of resolving two-way lake-atmosphere interactions (Fig. 3-b1). The improvement from GLARM-EA6 is also evident with the monthly surface air temperature over land where the bias of AOGCM-EA3 during Jan-Mar and Aug-Oct is nearly zero (Fig. 3-a2). The June and July bias, however, remains in both the AOGCM and GLARM simulations.

The added value of the GLARM simulations is also evident in the monthly precipitation. This is clearly reflected in the monthly climatology of the simulated precipitation where GLARM-EA6 drastically improved upon the GCM-EA3 monthly precipitation (Fig. 3-d2) . The large wet bias during Jan-Aug from the GCM-EA3 is significantly minimized by GLARM-EA6. Compared to GCM-EA3, GLARM-EA6 simulation was closer to the CRU data in nearly every month of the year. The mean bias of GLARM-EA6 is -0.07 mm/day as opposed to GCM-EA3 with 0.35 mm/day. Spatially, AOGCM-EA3 displays an abrupt increase in precipitation over the southern portion of the basin (Fig. 3-e2) whereas GLARM-EA6 simulates a gradual latitudinal gradient of precipitation similar to that in the CRU data (Fig. 3-d1, e1), leading to mostly smaller biases over the basin. The wet biases from AOGCM-EA3 near Lake Huron, Erie and Onatrio are noticeably reduced by GLARM-EA6 (Fig. 3-f1, f2).



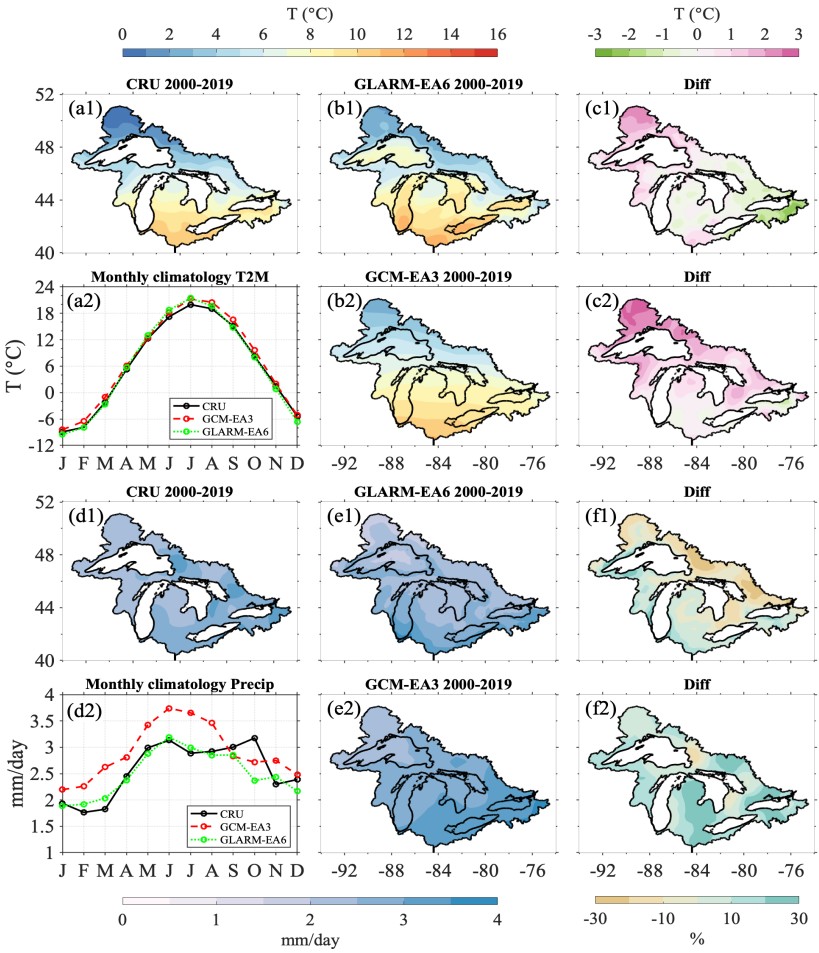

Figure 3: The climatology of surface air temperature and precipitation over the Great Lakes basin (2000-2019) from GLARM-EA6 simulation and GCM-EA3 simulation and their difference (model minus observations) relative to CRU land-based observations. Panels a2 and d2 show the monthly climatology of surface air temperature and precipitation over the land from 2000-2019.

Within the Great Lakes, LST and ice cover are the two most important physical lake variables that
influence the lake-atmosphere heat and water fluxes by affecting solar radiation, precipitation, and
evaporation, latent and sensible heat. Since the selected AOGCMs provide little or no representation
of the lakes, they are not included in the analysis. GLARM-EA6 and GLSEA LSTs show close
agreement with each other. LSTs vary significantly across the five lakes due to their immense
surface area, large geographic extent, and varying water depth. This spatial heterogeneity across





the lakes is primarily along the meridional direction, resulting in earlier warming in the southern lakes (Fig. 4-a,b,c). Temperature variations are the strongest during summertime when the northernmost, large, deep Lake Superior (average depth 147m) maintains a much cooler temperature of 12-14°C than the temperature of 22-24°C in the southernmost, small, shallow Lake Erie (average depth of 19 m). Additionally, GLARM-EA6 well captures the spatial heterogeneity within each lake. For example, GLARM reproduces the warmer eastern basin of Lake Superior during wintertime, the north-south temperature difference in Lakes Huron-Michigan during summertime, and the east-west thermal gradient in Ontario during fall.

In addition to resolving the spatial variability of climatological LST for each of the seasons, GLARM-EA6 performs well in reproducing the GLSEA lake-wide average LSTs (Fig. 5, a1-e1). The GLARM-EA6 predicted LSTs show close agreement with the GLSEA in both phase and magnitude for the five lakes. For example, the spring-early summer warming rate and the summer peaks are well reproduced by GLARM-EA6, which are often not well resolved in previous studies using 1D lake model coupled with RCMs (Bennington et al. 2014; Notaro et al. 2015). While GLARM-EA6 generally closely tracks GLSEA LST across the lakes, relatively large biases are simulated in the warming period in Lake Superior (June, July) and cooling period (October-December) in Lake Erie.

Although progress in ice modeling has been made, substantial challenges still remain and as a result larger biases than simulated LSTs typically exist (Anderson et al. 2018; Fujisaki et al. 2013, 2012). GLARM-EA6 captures the spatial variability of ice coverage observed in the GLICD ice data, with higher and lower ice coverage in shallow coastal and deep offshore regions, respectively(Fig. 4-e1, e2). GLARM-EA6 predicts ice cover fairly well in Lakes Michigan, Ontario, and Huron; however, it underestimates the magnitude of ice coverage in Lakes Superior and Erie (Fig. 5, a2-e2) although the observed values still fall in the ensemble envelopes. The shallowest lake, Lake Erie, is characterized by the highest ice coverage. GLARM-EA6 underestimates the Lake Erie ice cover by 15%-20% due to the warm biases of the winter LST. For the deepest lake, Lake Superior, GLARM-EA6 does not capture the highest ice coverage observed in March, but instead, it simulates a decrease in ice cover from February to March resulting in an 10% underestimate in ice cover.



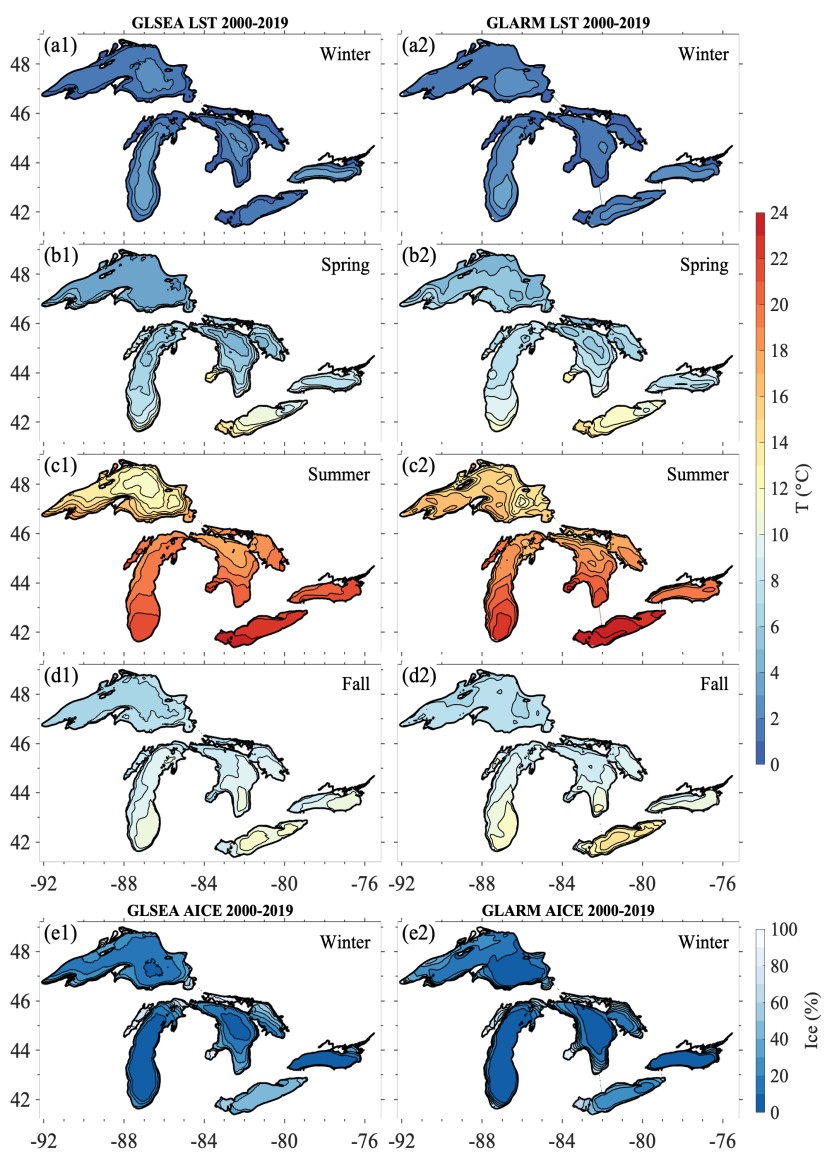

Figure 4: The LST seasonal climatologies (2000-2019) during (a1,a2) spring [April-June (AMJ)], (b1,b2) summer [July-September (JAS)], (c1,c2) fall [October-December (OND)], (d1,d2) winter [January-March (JFM)], and the ice cover climatologies (e1, e2). The GLSEA LST and GLICD ice observations are shown on the left panels; the GLARM-EA6 simulations are shown on the right panels.





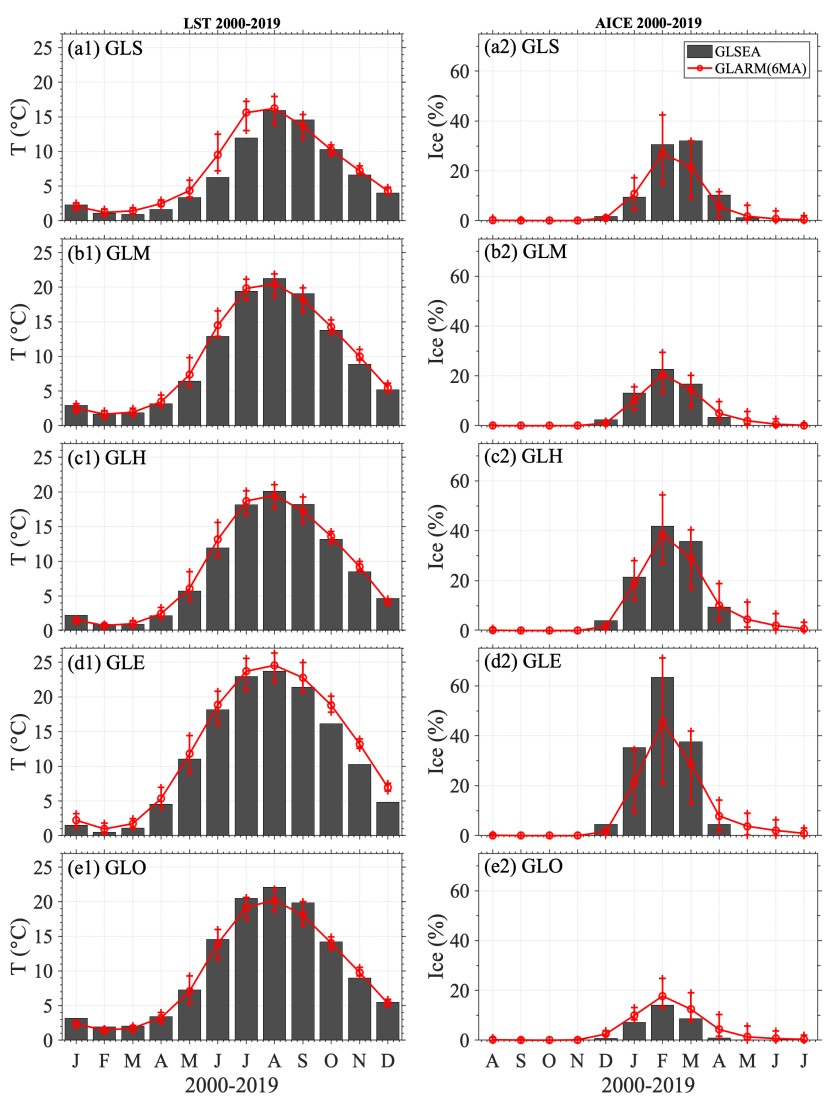

Figure 5: The monthly climatologies (2000-2019) of LST (left panels) and mean ice cover (right panels) in the five Great Lakes, respectively. The GLSEA LST and GLICD ice observations are shown in bar plots; the GLARM-EA6 simulations are shown in red lines with standard deviation of six GLARM configurations.

### 3.2.2 Projected Climate Change

**Surface Air Temperature**





Given the reliable performance of GLARM-EA6 in reproducing the present-day climate, we have increased confidence that GLARM is capable of making meaningful scenario-based projections of future climate. Here, we consider the RCP 4.5 and RCP 8.5 scenarios for the mid-century (2030-2049) and late-century (2080-2099) relative to the early twenty-first century (2000-2019). In the mid-century, the projected warming over the Great Lakes basin from two RCP scenarios is relatively similar, which is consistent with the IPCC (2013, 2021) report. The annual surface air temperature increases on average by 1.3°C in RCP 4.5 with a range of 0.8 to 1.9°C in six individual projections, and 1.7°C in RCP 8.5 with a range of 1.3 to 2.2°C by the mid-century (Fig. 6-a,c). The late century projected warming is much more substantial with 2.3°C warming in RCP 4.5 (1.8 to 2.7°C) and 4.4°C in RCP 8.5 (4.0 to 4.9°C) (Fig. 6-b,d). Spatially, all projections show a relatively higher increase by 0.1-0.5°C in the surface air temperature over land than over lake depending on the scenario and time frame considered, revealing the cooling effect of the lake. Such overlake and over-land temperature differences are most noticeable (4.0 vs. 4.5 °C) by the end of the century in the RCP8.5 scenario. In the mid-century, larger uncertainty in the projected surface air temperature, indicated by the standard deviation of the six-member ensemble projections, appeared in the northern region. In the late-century projections, the lowest (highest) uncertainties are found in the eastern part of the Great Lakes in RCP8.5 (RCP4.5) (Fig. 7).





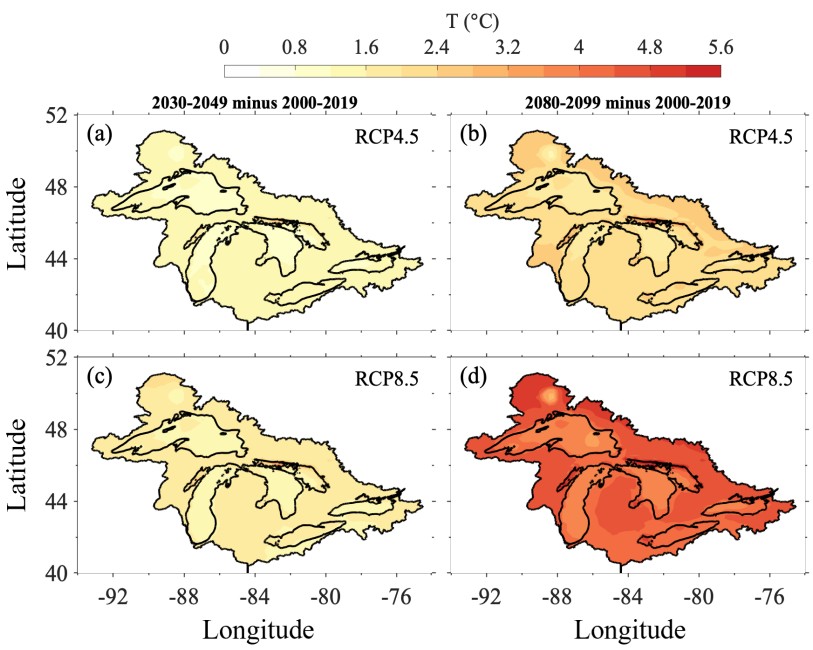

Figure 6: The changes in surface air temperature over the Great Lakes basin in the mid-century (2030-2049) and late-century (2080-2099) in RCP 4.5 and RCP 8.5 scenarios, predicted by GLARM-EA6.



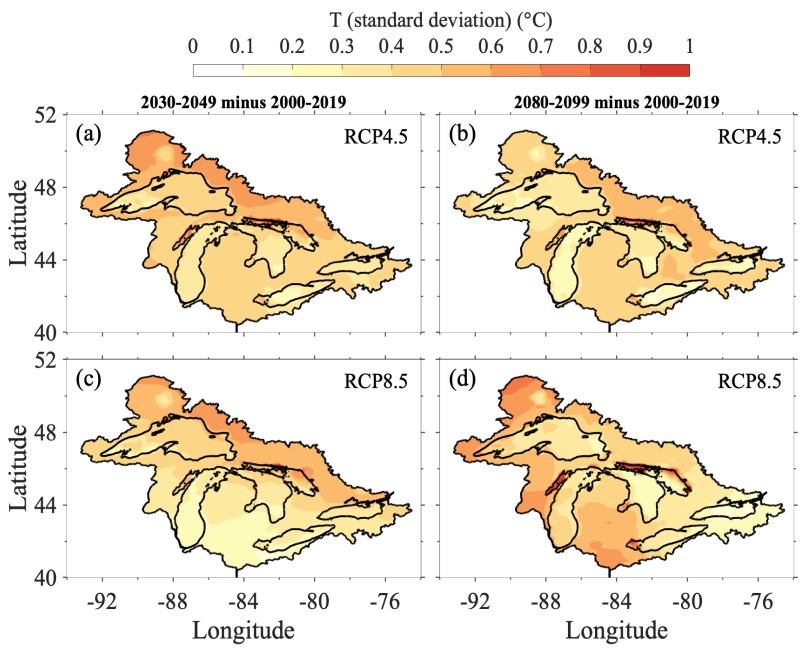

Figure 7: The uncertainties in GLARM-EA6 projected surface air temperature over the Great Lakes basin in the mid-century (2030-2049) and late-century (2080-2099) in RCP 4.5 and RCP 8.5 scenarios, indicated by the standard deviation of the six-member ensemble projections.

When considering monthly changes for each scenario and period averaged over the Great Lakes basin, increases in air temperature are predicted to be similar from April to October in each case (Fig. 8 and Table 3). More significant warming is projected during wintertime, which is particularly noticeable in the mid-century. A larger increase in temperature is projected for November and December for RCP 4.5 and December through March for RCP 8.5. By the end of the century, the temperature increases showed less seasonal variability. As summarized in the box-whisker plots of the six individual GLARM projections, the largest uncertainties across the six models in the projected warming are during the cold seasons (October through April) with variations of 2 to 3°C relative to the GLARM-EA6 ensemble mean, except for the late century in RCP 8.5 scenario when the largest uncertainties occur from July through October.



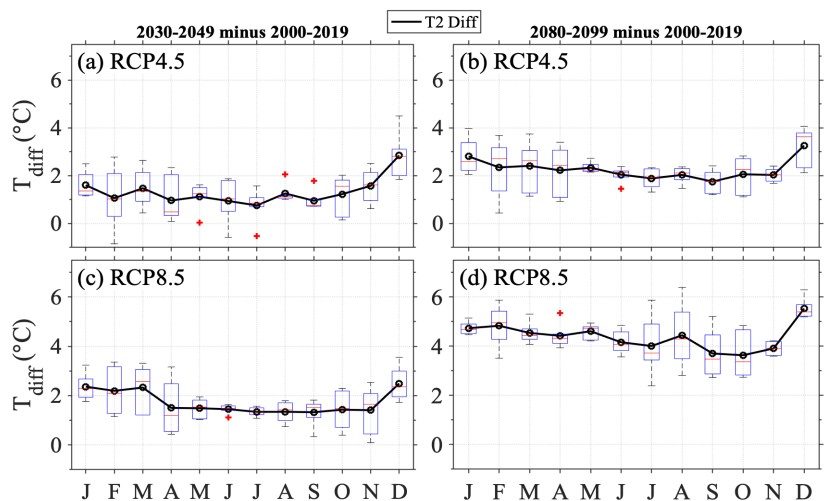

Figure 8: The average changes (black lines) in monthly surface air temperature over the Great Lakes basin in the mid-century (2030-2049) and late-century (2080-2099) in RCP 4.5 and RCP 8.5 scenarios, predicted by GLARM-EA6, uncertainties are indicated by the box-whisker plots based on from six individual GLARM projections.

## Precipitation

The enhanced warming as a result of the increased atmospheric GHGs, results in increased precipitation almost uniformly over the Great Lakes basin (Fig. 9 and Table 4). The projected mid-century increase is greater for RCP 4.5 (6%) than for RCP 8.5 (4%) despite the relatively similar atmospheric GHG concentrations over the period, confirming the lower degree of predictability of precipitation. However, by the end of the century, when the differences in GHG forcing are substantial, the precipitation increases are considerably greater for RCP 8.5 (18%) compared to RCP 4.5 (9%). The larger mid-21st century increase under RCP 4.5 and the substantial increase under RCP 8.5 during the latter half of the century align with the results presented in Wuebbles et al. (2019).

The spatial variation of the precipitation increase by the late 21st century is more pronounced under RCP 8.5 than RCP 4.5 (Fig. 9-b,d). Southern and western parts of the basin are projected to experience the biggest precipitation increases, up to 28% in RCP 8.5 and 15% in RCP 4.5. The uncertainties from GLARM precipitation projections show no clear spatial pattern, except for RCP 8.5 in which larger uncertainties are exhibited in the southwest region (Fig. 10). The standard deviation of total precipitation of the six-member ensemble predictions increases from near 0.3 mm/day at the northern parts of the basin to near 1 mm/day at the southern parts of the basin.





Table 3: The GLARM-EA6 projected changes in monthly, seasonal, and annual surface air temperature over land, lake, and the Great Lakes basin in the mid-century and late-century in RCP 4.5 and RCP 8.5 scenarios, relative to the present-day climate (2000-2019).

| | RCP4.5 | | | RCP4.5 | | | RCP8.5 | | | RCP8.5 | | |
| | 2030-2049 | | | 2080-2099 | | | 2030-2049 | | | 2080-2099 | | |
| | ΔT2 (°C) | | | ΔT2 (°C) | | | ΔT2 (°C) | | | ΔT2 (°C) | | |
| | Basin | Lake | Land | Basin | Lake | Land | Basin | Lake | Land | Basin | Lake | Land |
|---|---|---|---|---|---|---|---|---|---|---|---|---|
| Jan | 1.48 | 1.22 | 1.6 | 2.59 | 2.1 | 2.81 | 2.18 | 1.79 | 2.36 | 4.36 | 3.59 | 4.71 |
| Feb | 0.99 | 0.89 | 1.04 | 2.19 | 1.9 | 2.33 | 2.05 | 1.73 | 2.2 | 4.51 | 3.83 | 4.82 |
| Mar | 1.39 | 1.26 | 1.46 | 2.28 | 2.03 | 2.4 | 2.2 | 1.96 | 2.31 | 4.29 | 3.71 | 4.56 |
| Apr | 0.92 | 1.03 | 0.87 | 2.13 | 2.09 | 2.15 | 1.44 | 1.51 | 1.4 | 4.22 | 4 | 4.33 |
| May | 1.09 | 1.28 | 1.01 | 2.27 | 2.45 | 2.19 | 1.45 | 1.74 | 1.32 | 4.48 | 4.81 | 4.33 |
| Jun | 0.96 | 1.24 | 0.83 | 2.08 | 2.4 | 1.93 | 1.47 | 1.82 | 1.32 | 4.22 | 4.72 | 3.99 |
| Jul | 0.78 | 0.88 | 0.73 | 1.94 | 1.98 | 1.92 | 1.38 | 1.51 | 1.32 | 4.11 | 4.1 | 4.11 |
| Aug | 1.27 | 1.18 | 1.31 | 2.05 | 1.94 | 2.1 | 1.36 | 1.28 | 1.39 | 4.48 | 4.15 | 4.63 |
| Sep | 1.09 | 1.03 | 1.12 | 2 | 1.87 | 2.06 | 1.52 | 1.33 | 1.6 | 4.23 | 3.83 | 4.41 |
| Oct | 1.35 | 1.18 | 1.43 | 2.27 | 2.05 | 2.37 | 1.57 | 1.37 | 1.67 | 3.99 | 3.62 | 4.16 |
| Nov | 1.7 | 1.43 | 1.82 | 2.19 | 1.9 | 2.32 | 1.52 | 1.29 | 1.62 | 4.2 | 3.72 | 4.42 |
| Dec | 2.67 | 2.13 | 2.92 | 3.06 | 2.43 | 3.35 | 2.34 | 1.89 | 2.54 | 5.18 | 4.25 | 5.61 |
| JFM | 1.29 | 1.12 | 1.36 | 2.35 | 2.01 | 2.51 | 2.14 | 1.83 | 2.29 | 4.39 | 3.71 | 4.7 |
| AMJ | 0.99 | 1.18 | 0.9 | 2.16 | 2.31 | 2.09 | 1.45 | 1.69 | 1.35 | 4.31 | 4.51 | 4.22 |
| JAS | 1.05 | 1.03 | 1.05 | 2 | 1.93 | 2.03 | 1.42 | 1.37 | 1.44 | 4.27 | 4.03 | 4.38 |
| OND | 1.91 | 1.58 | 2.06 | 2.5 | 2.13 | 2.68 | 1.81 | 1.52 | 1.94 | 4.46 | 3.86 | 4.73 |
| Annual | 1.31 | 1.23 | 1.34 | 2.25 | 2.1 | 2.33 | 1.71 | 1.6 | 1.75 | 4.36 | 4.03 | 4.51 |



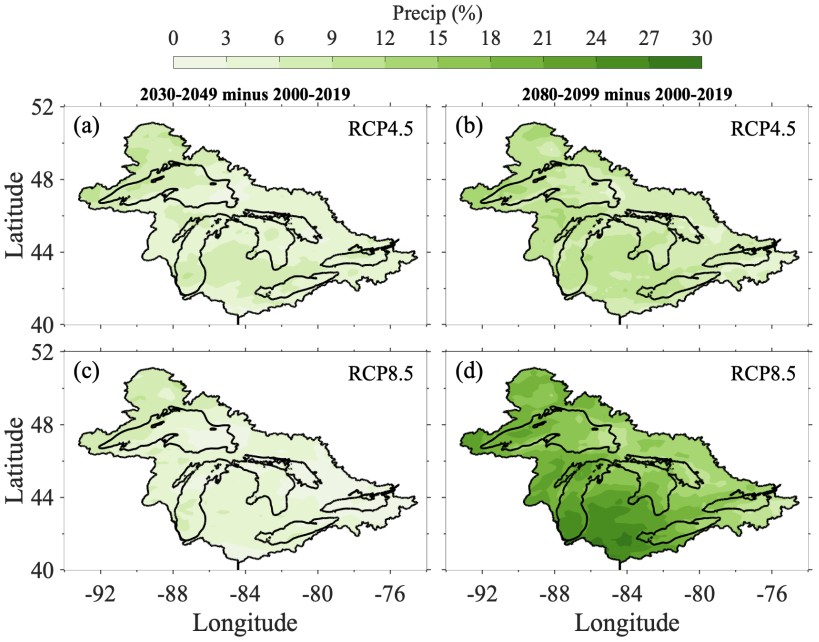

Figure 9: The project GLARM-EA6 changes in total precipitation over the Great Lakes basin in the mid-21st century (2030-2049) and late-21st century (2080-2099) in RCP 4.5 and RCP 8.5 scenarios.



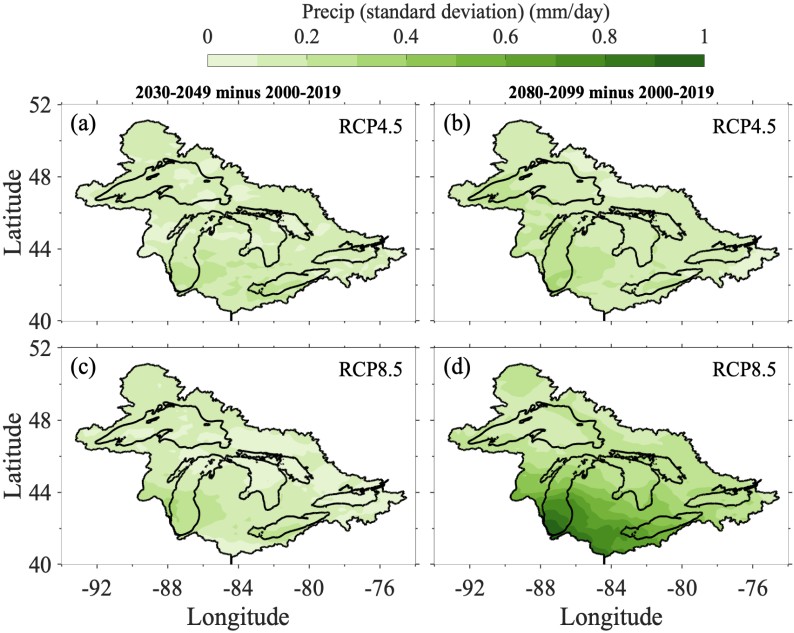

Figure 10: The uncertainties in GLARM-EA6 projected precipitation over the Great Lakes basin in the mid-century (2030-2049) and late century (2080-2099) in RCP 4.5 and RCP 8.5 scenarios, indicated by the standard deviation of the six-member ensemble projections.

Seasonally, while the GLARM-EA6 average shows basin-wide precipitation increases in nearly
all months, the predictions differ considerably between the individual six ensemble members (Fig.
11). The strongest and most robust signal is projected in spring, particularly in April and May,
which is found in all cases and is consistent with several previous studies (Byun and Hamlet 2018;
Notaro et al. 2015; Zhang et al. 2020). Not consistent with the aforementioned studies is that
GLARM-EA6 projects the enhanced spring precipitation persists into the summer at the end of the
century.



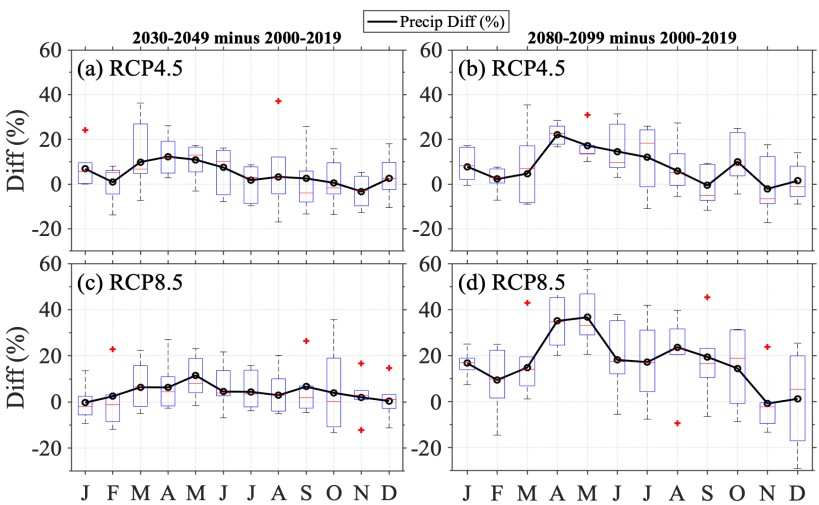

Figure 11: The average changes (black lines) in monthly surface precipitation over the Great Lakes basin in the mid-century (2030-2049) and late-century (2080-2099) in RCP 4.5 and RCP 8.5 scenarios, predicted by GLARM-EA6, uncertainties are indicated by the box-whisker plots based on from six individual GLARM projections.

**Lake Surface Temperature**

LST variability in each of the Great Lakes is significantly influenced by depth and geographic characteristics. The shallower lakes like Lake Erie exhibit larger seasonal LST variability than the deeper lakes like Lake Superior (e.g., summer LSTs are >25°C in Lake Erie and < 18°C in Lake Superior). Similar to the surface air temperature warming in the basin, the LSTs in the five lakes are projected to increase in time as the atmospheric GHGs accumulate (Table 5). The most significant LST increase occurs in Lake Superior under both RCP scenarios, followed by Lakes Michigan, Huron, Ontario, and Erie. Here we highlight the strong seasonal variability in lake warming as opposed to the seasonal pattern of surface air temperature increase (Fig. 12). In contrast to surface air temperature which shows little seasonal variability in its change, the LST increases in the lakes show substantial seasonal variability with the greatest changes projected in May and June in four of the five lakes. For example, the Lake Superior LSTs increase by 6.1°C and 3.2°C at the end of the century in RCP 8.5 and RCP 4.5, respectively, which are significantly larger than the annual mean respective increases of 4.1°C and 2.0°C (Fig. 12). As the summer progresses, the amplified warming begins to decline until the winter where it reaches its minimum increase of approximately 3°C in RCP 8.5 and 2°C in RCP 4.5 in the late-century. This is likely a result of some of the energy being used for ice melting and heat being transferred to the deepwater under unstratified conditions. Such patterns are projected across the lakes under all scenarios and for all periods, except for Lake





Table 4: The GLARM-EA6 projected changes in monthly, seasonal, and annual precipitation over land, lake, and the Great Lakes basin in the mid-century and late-century in RCP 4.5 and RCP 8.5 scenarios, relative to the present-day climate (2000-2019).

| | RCP4.5 | | | RCP4.5 | | | RCP8.5 | | | RCP8.5 | | |
| | 2030-2049 | | | 2080-2099 | | | 2030-2049 | | | 2080-2099 | | |
| | Δ P (%) | | | Δ P (%) | | | Δ P (%) | | | Δ P (%) | | |
| | Basin | Lake | Land | Basin | Lake | Land | Basin | Lake | Land | Basin | Lake | Land |
|---|---|---|---|---|---|---|---|---|---|---|---|---|
| Jan | 4.86 | 2.02 | 6.31 | 5.57 | 1.52 | 7.64 | -0.2 | -3.29 | 1.38 | 11.98 | 4.34 | 15.87 |
| Feb | 0.63 | -0.82 | 1.33 | 1.68 | -0.11 | 2.55 | 1.83 | -0.06 | 2.74 | 7 | 2.7 | 9.07 |
| Mar | 9.24 | 8.92 | 9.39 | 4.38 | 4.87 | 4.16 | 6 | 5.66 | 6.16 | 13.93 | 13.19 | 14.25 |
| Apr | 12.22 | 11.96 | 12.33 | 22.03 | 22.05 | 22.03 | 6.28 | 5.18 | 6.77 | 34.95 | 35.41 | 34.75 |
| May | 10.88 | 12.86 | 10.03 | 17.22 | 19.29 | 16.34 | 11.52 | 12.76 | 11 | 36.63 | 40.52 | 34.97 |
| Jun | 7.63 | 8.51 | 7.25 | 14.98 | 16.26 | 14.42 | 4.64 | 4.94 | 4.51 | 18.63 | 19.92 | 18.08 |
| Jul | 1.85 | 2.21 | 1.7 | 12.6 | 14.35 | 11.83 | 4.63 | 5.83 | 4.1 | 18.06 | 21.95 | 16.35 |
| Aug | 3.23 | 4.92 | 2.47 | 5.95 | 8.11 | 5 | 2.92 | 4.66 | 2.15 | 23.77 | 27.9 | 21.94 |
| Sep | 2.96 | 3.34 | 2.79 | -0.72 | 0.78 | -1.41 | 7.84 | 8.31 | 7.62 | 22.88 | 22.21 | 23.19 |
| Oct | 0.52 | 0.74 | 0.42 | 9.29 | 9.16 | 9.35 | 3.67 | 3.53 | 3.73 | 13.39 | 14.07 | 13.09 |
| Nov | 6.38 | 4.61 | 7.21 | 4.06 | 2.6 | 4.75 | -3.87 | -4.53 | -3.56 | 1.5 | -0.94 | 2.64 |
| Dec | 6.63 | 3.71 | 8.06 | 3.88 | 0.5 | 5.55 | 1.17 | -0.87 | 2.18 | 3.29 | -2.02 | 5.91 |
| JFM | 4.91 | 3.37 | 5.68 | 3.88 | 2.09 | 4.78 | 2.54 | 0.77 | 3.42 | 10.97 | 6.75 | 13.07 |
| AMJ | 10.24 | 11.11 | 9.87 | 18.08 | 19.2 | 17.59 | 7.48 | 7.63 | 7.42 | 30.07 | 31.95 | 29.26 |
| JAS | 2.68 | 3.49 | 2.32 | 5.94 | 7.75 | 5.14 | 5.13 | 6.27 | 4.62 | 21.57 | 24.02 | 20.49 |
| OND | 4.51 | 3.02 | 5.23 | 5.75 | 4.09 | 6.55 | 0.32 | -0.62 | 0.78 | 6.06 | 3.7 | 7.21 |
| Annual | 5.59 | 5.25 | 5.78 | 8.41 | 8.28 | 8.52 | 3.87 | 3.51 | 4.06 | 17.17 | 16.6 | 17.51 |





Erie which is projected to have the largest increase in summer. Spatially, the offshore waters where
depths are greatest are projected to experience the most significant warming across the lakes (Fig.
13).

Table 5: The GLARM-EA6 projected changes in annual LST in the five Great Lakes basins in the mid-century (2030-2049) and late century (2080-2099) in RCP 4.5 and RCP 8.5 scenarios, relative to the present-day climate (2000-2019).

|  |  | RCP4.5 | | | RCP4.5 | | | RCP8.5 | | | RCP8.5 | | |
|---|---|---|---|---|---|---|---|---|---|---|---|---|---|
|  |  | 2030-2049 | | | 2080-2099 | | | 2030-2049 | | | 2080-2099 | | |
|  |  | ΔLST(°C) | | | ΔLST(°C) | | | ΔLST(°C) | | | ΔLST(°C) | | |
|  |  | Min | Mean | Max | Min | Mean | Max | Min | Mean | Max | Min | Mean | Max |
| GLS |  | 0.87 | 1.16 | 1.52 | 1.77 | 1.97 | 2.38 | 1.18 | 1.56 | 2.11 | 3.96 | 4.11 | 4.53 |
| GLM |  | 0.79 | 1.12 | 1.51 | 1.66 | 1.86 | 2.21 | 1.21 | 1.51 | 1.95 | 3.71 | 3.98 | 4.57 |
| GLH |  | 0.75 | 0.99 | 1.3 | 1.55 | 1.77 | 2.04 | 1.02 | 1.33 | 1.72 | 3.48 | 3.66 | 4.15 |
| GLE |  | 0.51 | 0.81 | 1.07 | 1.08 | 1.37 | 1.52 | 0.56 | 0.95 | 1.16 | 2.4 | 2.73 | 3.02 |
| GLO |  | 0.89 | 1.15 | 1.5 | 1.8 | 2.03 | 2.27 | 1.18 | 1.45 | 1.93 | 3.96 | 4.15 | 4.44 |

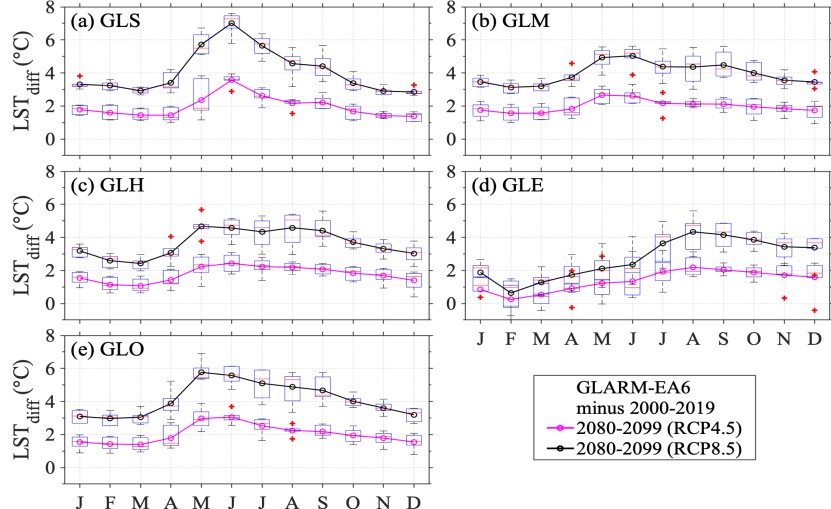

Figure 12: The average changes (black and purple lines) in LSTs over the five Great Lakes in the late-century (2080-2099) in RCP 4.5 and RCP 8.5 scenarios, predicted by GLARM-EA6, uncertainties are indicated by the box-whisker plots based on the six-member ensemble projections.

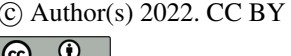

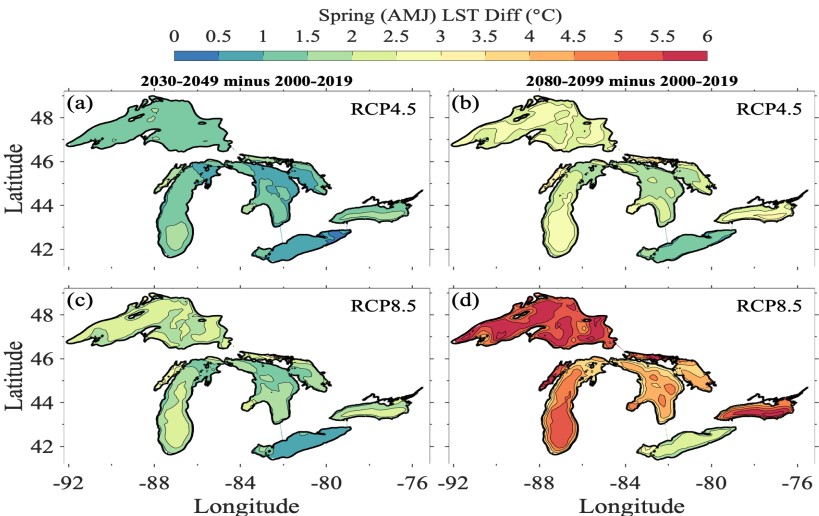

Figure 13: The changes in spring (AMJ) LSTs over the Great Lakes basin in the mid-century (2030-2049) and late-century (2080-2099) in RCP 4.5 and RCP 8.5 scenarios, predicted by GLARM-EA6.

**Lake Ice**

In the winter, the warming signals are reflected in an overall reduction in ice coverage and duration (Fig. 14) in all scenarios and periods. Here we present the projected lake conditions in the late-century as an example (Fig. 14). The ice cover projections show the least uncertainty in RCP 8.5 scenario in the late-century, in response to the strongest warming. In the RCP 8.5 scenario, mean ice coverage in February is projected to reduce to between 3% and 6% across the lakes. This indicates that ice cover percentage in the five lakes will become more uniform compared to the present-day conditions (Fig. 5). The ice duration (defined with a threshold of 10% ice coverage at a given model grid) is projected to decrease correspondingly (Fig. 15). By the mid-21st century, the ice duration is projected to decrease by 5 to 25 days depending on the scenario and location; and by the late century up to 50 days in the coastal regions where higher ice covers are typical in the present-day climate.




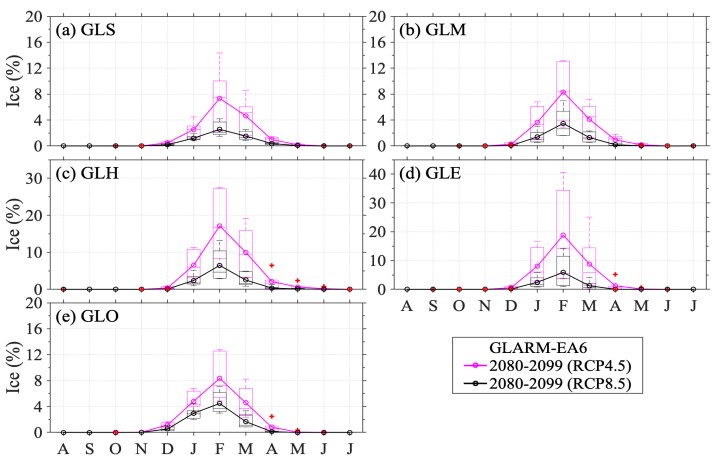

Figure 14: The projected monthly mean ice covers in the five Great Lakes in the late century (2080-2099) in RCP 4.5 and RCP 8.5 scenarios, predicted by GLARM-EA6, uncertainties are indicated by the box-whisker plots based on the six-member ensemble projections.

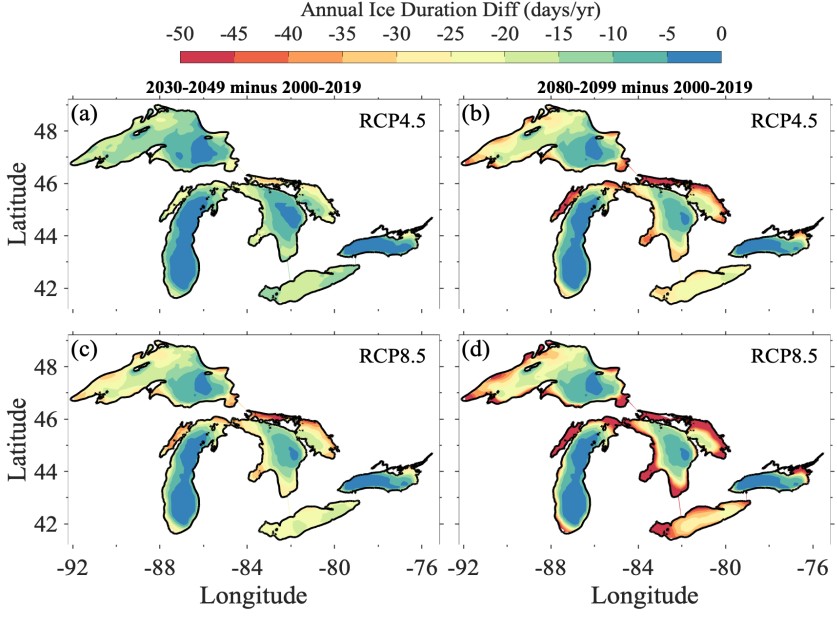

Figure 15: The reduction in ice duration (days) in the Great Lakes in the mid-century (2030-2049) and late-century (2080-2099) in RCP 4.5 and RCP 8.5 scenarios, predicted by GLARM-EA6.



## 4 Discussion and Conclusions

### 4.1 Model Advancement

The Great Lakes are a key element in regional climate of the basin and play an important role in influencing local weather patterns and climate processes. Climate processes are changing, accompanied by changes in the Great Lakes. Many of these complex changes are regulated by interactions among the atmosphere, lake, ice, and surrounding land areas that can also have an important influence in regulating regional climate. The lack of fully integrated regional models that resolve 3-D lake dynamics may result in inaccurate projections of climate change for the basin and associated adaptation and mitigation measures. To the best of our knowledge, this study presents the first climate change projections including both the Great Lakes basin and the changes in the five Great Lakes that has employed a two-way coupled regional climate model with a 3-D lake model (i.e. GLARM).

Using the three carefully selected CMIP5 AOGCMS and two domains (large continental and small regional), we show that the GLARM six-member ensemble average (GLARM-EA6) substantially reduces the surface air temperature and precipitation biases of the driving AOGCM ensemble average. The improvements are not only displayed from the atmospheric perspective but also include lake surface temperature and ice coverage and duration.

### 4.2 Summary of Climate Projections

The GLARM climate change projections are performed for the mid-century (2030-2049) and late-century (2080-2099) for the RCP 8.5 "business as usual" scenario and the RCP 4.5 moderate mitigation scenario. The surface air temperature over the Great Lakes Basin is projected to increase in all months regardless of the scenario, period of consideration and ensemble member. Under RCP 8.5, the Great Lakes basin is projected to warm by 1.3-2.2°C by the mid-21st century and 4.0-4.9°C by the end of the century relative to the early-century (2000-2019). Moderate mitigation (RCP 4.5) reduces the mid-century warming to 0.8-1.9°C and late-century warming to 1.8-2.7°C. The largest amount of warming is projected during the winter, consistent with the predictions from Byun and Hamlet (2018) and Zhang et al. (2020). Since previous studies consider different time periods and GHG emissions scenarios for their projections, a comparison of precise magnitude of changes is not possible; nevertheless qualitative comparisons can be made. The GLARM simulations presented here project surface air temperature increases slightly smaller than those of previous studies (e.g., Notaro et al. 2015; Zhang et al. 2020). For example, by 2080-2099 under RCP 8.5, Notaro et al. (2015) project annual overland air temperature to increase by up to 5.9°C relative to 1980-1999, while GLARM predicts an increase of 4.5°C relative to 2000-2019. When considering that the CRU data show a 0.5°C difference between the baseline periods of the two studies, the GLARM RCP 8.5 ensemble projects a reduction by about 0.9°C compared to Notaro et al. (2015).





As for the spatial variation of the predicted increase, GLARM's relatively larger increase in the northern part of the basin (particularly under RCP 4.5 by the end of the 21st century) agrees with Xiao et al. (2018).

Annual precipitation in GLARM is projected to increase for the entire basin with the largest relative increases in spring and early summer when current precipitation is highest and little increase in winter when it is lowest. There is some consensus among previous studies at the annual timescale, However, these studies project decreases in summer and increases in winter and spring (e.g., Byun and Hamlet 2018; Notaro et al. 2015; Zhang et al. 2020). In addition, the smaller Great Lakes domain configuration projects a wider range of precipitation suggesting that the dynamics over the Great Lakes region are more constrained by the lateral boundary conditions and inherit precipitation patterns from the driving AOGCMs. This is particularly evident for the MPI-ECM-MR downscaling cases where the projected increases are relatively large with the smaller GLARM domain and muted changes with the larger domain. This reinforces the use of two different modeling domains  The large North America domain to account for both dynamic consistency of climate processes resolved in the GLARM and allow the regional scale feature to fully develop; Meanwhile, the small domain GLARM, similar to other RCM configuration for the Great Lakes climate study to represent the uncertainty inherited from different GCMs and enhance computational efficiency.

LST also increases across the five lakes in all of the simulations, but with a stronger seasonal signature compared to surface air temperature which was relatively constant in all months. The strongest warming was projected in spring followed by strong summer warming suggesting earlier and more intense stratification in the future. In contrast, a relatively small increase in fall and winter LST is projected with a minimal increase with heat transfer to the deepwater due to strong mixing and energy required for ice melting. Correspondingly, GLARM ensemble projects decreased ice cover and duration. Of particular note, the highest monthly mean ice cover is projected to be only 3 to 6% across the lakes by the end of the 21st century in RCP 8.5; and ice duration will decrease by up to 30- 50 days in the coastal regions. The few climate change studies that dynamically downscale Great lake temperatures and ice cover used 1-D lake models embedded in the RCMs (Notaro et al. 2015; Xiao et al. 2018). The GLARM simulations are consistent with these previous studies, however, the magnitude of the increase is considerably less than Xiao et al. (2018) who project increases of 3.5 to 4.0 °C for 2070-2100 relative to 1975-2005 under RCP 4.5 and Notaro et al. (2015) who project increases of up to 8°C by 2080-2099 relative to 1980-1999 under RCP 8.5. Counterintuitively, both of these studies project larger ice coverage than the GLARM's simulation. It should be noted that their ice coverage simulations were heavily limited by their 1D lake-ice model; both studies explicitly noted that the absence of 3D model produced substantial summer warm biases and cold biases in winter (Notaro et al. 2015) with earlier ice onset and excessive mid-winter ice (Xiao et al. 2018). Hence, the 3D representation of lake and ice processes within GLARM could feedback to dampen changes in lake warming and ice coverage and duration.





Collectively, the projected changes in the atmosphere and the lakes are expected to modify weather and climate extremes and associated coastal hazards, including extended local heat stresses and marine (lake) heatwaves, heavy precipitation, rising lake levels, and coastal flooding (Huang et al. 2021a,b; Notaro et al. 2021; Wuebbles et al. 2019; Zhang et al. 2019). With unabated GHG gas emissions, all lakes will experience less ice coverage extent and duration and even ice-free winters. This will significantly alter the overlake heat and moisture fluxes during the cold season, which could lead to intensified winter storms. For example, the increased winter moisture supply from the lakes along with events of cold air mass (e.g. polar vortex) can create ideal conditions stronger lake effect snowfall events (Basile et al. 2017; d'Orgeville et al. 2014). As such, we advocate that a regional earth system modeling system with integration of observing networks becomes vitally essential to guide decision-makers in response to climate change and climate-driven coastal hazards.

**Author Contributions:** Conceptualization, P.X.; methodology, P.X., X. Y, C. H., J.S.P.; software, P.X., X.Y, C. H.; validation, C.H., X.Y.; visualization, C.H., formal analysis, P.X., J.S.P., P.Y.C., X.Y., M.B.K; resources, P.X.; writing—original draft preparation, P.X., M.B.K.; J.S.P.; writing—review and editing, P.X., J.S.P., P.Y.C.; supervision, P.X.,; project administration, P.X.; funding acquisition, P.X. All authors have read and agreed to the published version of the manuscript.

**Funding:** This research was partly supported by the National Aeronautics and Space Administration, Grant 80NSSC17K0287. Hydrodynamic modeling was also supported, in part, by COMPASS-GLM, a multi-institutional project supported by the U.S. Department of Energy, Office of Science, Office of Biological and Environmental Research, Earth and Environmental Systems Modeling program.

**Data Availability:** The Great Lakes Surface Environmental Analysis (GLSEA) is available from `https://coastwatch.glerl.noaa.gov/glsea/glsea.html`. The Great Lakes Ice Cover Database (GLICD) is available from `https://www.glerl.noaa.gov/data/ice/#historical`. The RegCM4 code is available through`https://www.ictp.it/research/esp/models/regcm4.aspx`. The CRU data is available from `https://crudata.uea.ac.uk/cru/data/hrg/#current` The FVCOM code is available through `http://fvcom.smast.umassd.edu/fvcom/`. Further inquiries can be directed to the corresponding author.

**Acknowledgments:** This is the contribution XX of the Great Lakes Research Center at Michigan Technological University. The Michigan Tech high-performance computing cluster, *Superior*, was used in obtaining the hydrodynamic modeling results presented in this publication.

**Conflicts of Interest:** The authors declare no conflict of interest.





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
