# Peer review of "Climate Projections over the Great Lakes Region: Using Two-way Coupling of a Regional Climate Model with a 3-D Lake Model"

_Geoscientific Model Development, 2021_

## Referee Comment (RC3)

**Review of "Climate Projections over the Great Lakes Region: Using Two-way Coupling of a Regional Climate Model with a 3-D Lake Model", by Xue et al.**

In this study, the authors present and analyze climate projections over the Laurentian Great Lakes regions using an RCM coupled to a 3D lake model. The authors discover that the model setup substantially reduces model biases relative to the driving GCMs, and that future GHG emissions may lead to substantial changes in the near-surface climate in the region.

The paper uses model simulations to assess climate change in a robust way (given computational constraints) over a region where many people live and depend on the lakes for their livelihood. The selection of the driving GCM using an objective method is appreciated, though I'm not sure a penalty for distance from the ensemble mean is appropriate. Moreover, the manuscript is well written, and the figures are generally clear. Also, the abstract and introduction reads very well.

This study thus overall demonstrates the potential to make a substantial contribution to the scientific literature. However, I have some concerns, which require minor revisions of the manuscript. In general, I could recommend publication of this study if the comments specified below are sufficiently addressed.

**General Comments**

1. As a general comment, there seem to be several inconsistencies in the naming of experiments, evaluation products, variable names etc. throughout the manuscript, figure labels and figure captions. See specific comments for examples, but please carefully check the entire manuscript for other inconsistencies.

2. Difference between using the small and large domain is only briefly discussed for precipitation, though I feel this is important information to share. Does model performance and/or climate sensitivity differ between the two domains? Or do both domains yield very similar results (e.g., for T and LST)? And which domain approach do the authors recommend for future research in this region? Such information could be covered in the discussion section.

**Specific comments**

1. L1: No need to reply to this comment, but I am a little surprised that the authors choose GMD as a journal to publish their work. Given that the focus of the study is on the results of the future projections, I believe that a content-journals like for example ESD could have been a better fit for this work. That said, I respect the author's choice of GMD and do not suggest transferring this manuscript to a different journal.

2.   L51: does this statement refer to an area or temporal change? Please clarify.

3.   L79-82: In this context, it could be interesting to check what the recent scientific results obtained as part of the ISIMIP lake sector tell for the Laurentian Great Lakes. See https://www.isimip.org/outcomes/publications-overview-page/

4.   L109: Importantly, RCP8.5 is not to be considered 'business as usual', but a 'high-end emission scenario'. And I suggest referring to RCP4.5 as a 'moderate mitigation scenario'

5.   L160-162: please add one or more refs to back this statement.

6.   Table 2: I suggest marking the selected GCMs in bold in this table

7.   Figure 3: caption and figure labels say 'GCM' but manuscript says 'AOGCM'. Please make this consistent (I think GCM is used more often nowadays).

8.   Figure 4: Are the wintertime LSTs water temperatures taken only during the ice-free period or the average of the whole season (i.e., combined snow/ice/open water)? Please clarify. Also, caption says 'GLICD' but title of panel e says GLSEA. Also, has the acronym AICE (title panels e1-2) been introduced?

9.   Figure 5: legend: GLARM(6MA), while text and other figures use GLARM-EA6. Also, spell out lake names (acronyms are not introduced in paper and they add no value, see also figure 12 and elsewhere)

10.  Figure 7 & 10: is this the standard deviation of the change (future minus past) or of the future state? Please clarify in the manuscript and/or caption

11.  L327-328 & L332-333: I wonder if these changes (4 and 6%) are sufficiently different to say that RCP4.5 gives a stronger wetting than RCP8.5 for mid-century. Probably the uncertainty bands are largely overlapping? In that case I would rather say that they project a similar wetting.

12.  Fig. 14: to better understand the change, it would be more useful to also plot the present-day ice cover, or to plot the change in ice cover (future – present)

**Textual comments**

1.  L26, 'are' > 'is'.

2.  L61: 'will' > 'could'.

3.  L72: 'predicted' > 'projected' (always use projections in the context of future climate) and 'atmospheric greenhouse gasses (GHGs)' > 'greenhouse gas (GHG) emissions'.

4.  L89 and elsewhere: I'd suggest specifying 'Great Lakes' to 'Laurentian Great Lakes' throughout the manuscript, to avoid confusion with the African Great Lakes.

5.  L101: FVCOM, this acronym hasn't been introduced yet

6.  L106: 'a RCM' > 'an RCM'

7.  L116: 'LakesAtmosphere' > 'Lakes Atmosphere'

8.  L116: 'iceatmosphere' > 'ice-atmosphere'

9.  L155: 'projections' > 'assessment reports'

10. L233: 'predictions' > 'projections' (check elsewhere, predicted>projected)

11. L252: remove space before '.' Check elsewhere for double or missing spaces.

12. L306: 'much more substantial' > 'more pronounced'

13. L309: 'cooling' > 'buffering'

14. L310: 'overlake' > 'over-lake'

15. L429: check punctuation

16. L428: check sentence (drop 'changes'?)

---

## Author Comment (AC2)

**Reviewer2:**

RC2: **'Comment on gmd-2021-440'**, Anonymous Referee #2, 06 Feb 2022

This study demonstrates the use of a two-way coupling of regional climate models with a 3-D hydrodynamic model (GLARM) of the US Great Lakes based on three selected CMIP5 AOGCMS and two spatial domains. The authors first evaluate the degree of skill of the models and then examine two climate scenarios (RCP 4.5, 8.5) and evaluate their impact on the Great Lakes Basin during the mid and late 21st century. They show the spatial and temporal variability in expected precipitation, ice cover and LST for all the great lakes.

General

I found the paper to be very well written and timely. It represents one of the only cases (if not the only one) in which two-way coupling of a lake 3D hydrodynamic model and regional climate model have been use to examine the potential impacts of project climate change under various climate scenarios.

Nevertheless a number of questions come to mind. Were the lake models driven with inflows and outflows and do the models account in any way the likely increase in inflows especially during the rainy spring period as projected by the results of simulations? If the inflows and outflows are neglected in the simulation, this should be mentioned and discussed as I assume they will have an impact on water temperature.

Response: Thanks for the question! The hydrological cycle is not simulated in this paper for two reasons. First, surface hydrology requires great effort and needs to be studied separately. The water levels of the Great Lakes are primarily governed by the net basin supplies (NBS) of each lake (which are the sum of over-lake precipitation and basin runoff, and minus lake evaporation), in a combination with the Great Lakes regulation plan as well as inter-lake flows to describe a complete water budget. This requires a suite of models to be properly integrated to project water level changes. In fact, we have done it in our recent study of the Great Lakes water level, which has been submitted to *Journal of Hydrology*, "Future Rise of the Great Lakes Water Levels under Climate Change" by Miraj B. Kayastha, Xinyu Ye, Chenfu Huang, Pengfei Xue* (corresponding author) (in revision). In that paper, we integrated GLRM (for over lake precipitation, evaporation), LBRM(Large Basin Runoff Model for river runoff into each lake ), CGLRRM (Coordinated Great Lakes Regulation and Routing Model for inter-lake flow and regulation plans) to project the changes in surface hydrology and the Great Lakes water level change in the future. Given the complexity and importance of this topic, it is beyond the scope of this study. Second, the water level fluctuation (1-2 m) does not impact the surface area of the Great Lakes (considering the depth and size of these lakes); therefore, water level change (which is certianly critical for coastal erosion, navigation) does not play an important role in influencing lake-air heat fluxes and climate change, that's why we simulate the over lake evaporation (latent heat flux) but did not simulate complete surface hydrological cycle in this study. In addition, the

heat transport between lakes associated with inter-lake flows is secondary. It falls in the uncertainty of surface heat fluxes (i.e. the primary cause of lake thermal change) in the GLARM.

These are now explicitly mentioned in the discussion section as "*We note that this study does not directly simulate the surface hydrological cycle for three reasons. First, the water levels of the Great Lakes are primarily governed by the net basin supply (NBS) of each lake (over-lake precipitation, river runoff, and lake evaporation), in combination with natural and regulated inter-lake flows. The projection of water level changes requires the integration of a suite of models. Such integration is documented in our separate study, in which we use GLARM (for over-lake precipitation, lake evaporation), LBRM (Large Basin Runoff Model) for river runoffs into each lake, CGLRRM (Coordinated Great Lakes Regulation and Routing Mode) for inter-lake flows. Given the complexity of the projection of the surface hydrological cycle, it is beyond the scope of this study. Second, the impact of water level change on the surface area of the Great Lakes is negligible; therefore, water level change does not play a critical role in influencing lake-air heat fluxes and climate change. Third, compared to the primary factor (surface heat fluxes) of lake thermal change, the heat transport between lakes associated with inter-lake flows is secondary on the lake basin-wide scale. It falls in the uncertainty of surface heat fluxes in the GLARM projections.*"

The authors mention two key physical processes in the lakes but don't present any data or model output for the two processes. The first is the possible change in stratification which as the authors quite correctly point out in the introduction can greatly impact the ecosystem. The second is the mention of the possible effect of the mixing of heat from the surface to bottom. It would be very interesting to see what the projected change in the duration in stratification is expected to be (as expected by the authors, lines 436-437) and whether there is any clear increase in bottom water temperature to support the mechanism suggested by the authors (e.g. line 437-439). I would also have liked to see a brief discussion as to the quality of the 3d lake model results in relation to other models and where the weaknesses may be.

Response: Good question! This question was asked by other reviewers too! In the revised version, one of the major changes we made is to address this question. We have dedicated a thorough discussion from lines 279-302 and new figure 12 (projected thermal structure change) to this. Here I try to make a short summary: It is related to the strong early stratification in the deep lakes that cause a significant increase in spring LST. And the higher ice cover in Lake Erie (which leads to a relatively lower increase in LST during winter), and relatively lower ice in deep lakes. This is because deep lakes are, by nature, large heat reservoirs that can transfer heat from a deep lake layer to the surface to reduce ice formation. The best example is the observed ice coverage of the shallowest lake (Erie) and the second deepest lake (Ontario). Both lakes have small surface areas but

significantly different water depths (mean water depths are 19 m and 86 m, respectively, Fig. 1, panel b), resulting in high (low) winter ice cover in Lake Erie (Ontario) (Fig. 4).

Here is new figure 12 (screenshot) for the projected change in the duration in stratification is expected to be and clear increase in bottom water temperature to support the mechanism suggested by the authors.

[Figure]

**Figure 12.** The lake thermal structures in the central Lake Superior (upper panel) and Lake Erie (middle panel) in the present-day climate (2000-2019) and the late century (2080-2099). Bottom panel:The comparison of projected changes in monthly LST in Lakes Superior and Erie in the late century (2080-2099) in RCP 8.5, relative to the present-day climate (2000-2019).

What are the important 3-D lake processes that 1-D lake models fail to resolve and impact lake thermal structure and ice cover? We have a separate manuscript (*Importance of Coupling a 3D Lake Model to the Regional Climate Model in Simulating the Great Lakes System by Xue P, Notaro M., Huang C., Zhong Y., Peters-Lidard, C., Cruz, C., Kemp, E., Kristovich, D., Kulie, M., Wang, J., Huang, C., Vavrus, S* to be submitted to Journal of Hydrometeorology, which specifically addresses this. As you may notice, this is a study with a separate group of collaborators and we don't steal the main message from that study. So

we prefer not to discuss it in this manuscript (thank you for your understanding). However, We are happy to share our findings here below.

We have done a set of process-oriented numerical experiments in the aforementioned study and results show that the most important lake process that impact LST and ice is the turbulent mixing process, which is controlled by turbulent kinetic energy calculated by shear production, buoyancy production, rate of dissipation, and advective and turbulent transport. Many of these processes require 3-D fields to be correctly estimated. In the 1-D model, these estimations have to be simplified with 2 m wind speed, the Brunt-Väisälä frequency, the latitude-dependent Ekman decay, and often with an empirical modification factor to find a lumped eddy diffusivity coefficient. This is the most important process that the 3-D lake model outperforms 1-D lake models in simulating lake thermal structure.

Also, our previous publication (Ye, X., Anderson, E. J., Chu, P. Y., Huang, C., & *Xue, P.[Corresponding author] (2018). Impact of Water Mixing and Ice Formation on the Warming of Lake Superior: A Model-guided Mechanism Study. Limnology and Oceanography) studies the impact of strong and weak winter mixing on the lake heat content and lake surface temperature and ice. During wintertime, stronger mixing causes a warm surface layer by allowing the heat transport to the surface from the warmer deep layer, causing less ice and stronger evaporation. Weaker mixing results in a strong winter stratification, and cold surface with extensive ice cover.

There are other important 3-D processes that can only be resolved in 3-D lake model, including heat transport associated with large-scale circulation, and density-driven two-layer baroclince flow, upwelling, and ice drifting, which significantly affect the spatial pattern ice coverage (we have done simulations with and without ice drifting).

Specific comments:

Line 28- % of what

Response: the sentence is revised as "Correspondingly, the highest monthly mean ice cover is projected to be reduced to 3-15\% and 10-40\% across the lakes by the end of the century in RCP 8.5 and RCP 4.5, respectively."

Line 97- remove the word in

Response: removed.

Line 101- FVCOM not yet defined

Response: Corrected. Finite Volume Community Ocean Model (FVCOM)

Line 116- add space between ice and atmosphere

Response: added.

Line 145-149- how accurate are these data compared to actual measurements? Is a correction required before using to validate the model?

These data have been used in many studies in the Great Lakes region. It is appropriate to use the dataset for a basin-wide assessment (but not the best choice for site-specific validation if in-situ data is available). The GLSEA LST has a very good representation of the LST spatial pattern (based on our studies with three NOAA-funded data assimilation projects for the Great Lakes); however the GLSEA LST data quality is much lower during the wintertime because of the quality and availability of satellite data and ice cover. That's why we focused on ice cover data to validate the model performance during winter.

Eq 1- over what time and spatial resolution is this calculated? Is there a reference to the use of this type of equation?

Response: We conducted the model reliability analysis using model-simulated NA-averaged temperature in the historical periods (1901-2005) and the future period (2006-2100) in RCP 8.5 scenario. The three GCMs with the highest reliability scores are selected to drive GLARM for the present-day and two future periods in each scenario.

As we cited in our paper, this method including the equation is documented in (Giorgi and Mearns, 2002, Journal of Climate: Calculation of Average, Uncertainty Range, and Reliability of Regional Climate Changes from AOGCM Simulations via the "Reliability Ensemble Averaging" (REA) Method). This is equation (4) in their paper.

The equation is also shown as equation 2 in Miao, C., Duan, Q., Sun, Q., Huang, Y., Kong, D., Yang, T., Ye, A., Di, Z., and Gong, W.: Assessment of CMIP5 climate models and projected temperature changes over Northern Eurasia, Environmental Research Letters, 9, 055 007, 2014. (this citation is also added in the revision)

After the three GCMs are selected using reliability analysis, we then used taylor diagrams (RMSD, correlation, Std; figure 2 upper panel) and warming trend analysis (figure 2, lower panel ) to check (validate) if our GCM selection is appropriate. The four statistic metrics are for independent validation of our GCM selection.

Table 2- change RMSE in the legend to RMSD

Response: corrected.

Figure 3- would be nice to have lake names on this and the other figures, especially for those not familiar with the Great Lakes.

Response: we've added lake names to all applicable figures.

Figure 4- legend- there is a mistake in the seasons and figures. A1,a2 for example are the winter and not spring.

Response: Corrected.

Figure 8 legend- word missing from last line.

Response: Corrected.

Table 3- what is ΔT2 in column title?

Response: We replaced ΔT2 with "T2 change", along with more clear table title: The projected changes in monthly, seasonal, and annual surface air temperature over land, lake, and the Great Lakes basin in the mid-century and the late century in RCP 4.5 and RCP 8.5 scenarios, relative to the present-day climate (2000-2019).

Line 343- "particularly in Aril and May" this is correct only for the end of century results

Response: corrected.

Line 361- Figure 12 should be Table 5

Response: corrected.

Citation: https://doi.org/10.5194/gmd-2021-440-RC2
A note:
Finally, we want to let you know that in response to other reviewer's question and our (Co-PI) internal discussion (also in consulting with a senior climate scientist at MIT) on the concern of whether or not (and how) we should combine these 3 GLARM-large domain model results and 3 GLARM-small domain model results. We agreed that a simple ensemble average seems questionable because these results are from two sampling groups that can possess different uncertainty distributions. We decided just to use one of the domains. We selected the small domain GLARM, which is similar to other RCM configurations for the Great Lakes climate studies, to represent the uncertainty inherited from different GCMs and enhance the computational efficiency. Nonetheless, please note that the results (GLARM-EA3) are similar to our previous 6-member ensemble results (GLARM-EA6), and all conclusions remain unchanged. Still, we did update the results (like numbers and tables) throughout the manuscript.

Thank you again for your questions and suggestions. I hope we have addressed your concerns and questions satisfactorily.

---

## Author Comment (AC4)

**Reviewer3:**

Review of "Climate Projections over the Great Lakes Region: Using Twoway Coupling of a Regional Climate Model with a 3-D Lake Model", by Xue et al.

In this study, the authors present and analyze climate projections over the Laurentian Great Lakes regions using an RCM coupled to a 3D lake model. The authors discover that the model setup substantially reduces model biases relative to the driving GCMs, and that future GHG emissions may lead to substantial changes in the near-surface climate in the region. The paper uses model simulations to assess climate change in a robust way (given computational constraints) over a region where many people live and depend on the lakes for their livelihood. The selection of the driving GCM using an objective method is appreciated, though I'm not sure a penalty for distance from the ensemble mean is appropriate. Moreover, the manuscript is well written, and the figures are generally clear. Also, the abstract and introduction reads very well. This study thus overall demonstrates the potential to make a substantial contribution to the scientific literature. However, I have some concerns, which require minor revisions of the manuscript. In general, I could recommend publication of this study if the comments specified below are sufficiently addressed.

General Comments

1. As a general comment, there seem to be several inconsistencies in the naming of experiments, evaluation products, variable names etc. throughout the manuscript, figure labels and figure captions. See specific comments for examples, but please carefully check the entire manuscript for other inconsistencies.

Response: Thanks! We've gone through the paper in the revision and ensured the consistency of experiments, names, labels, captions, etc.

2. Difference between using the small and large domain is only briefly discussed for precipitation, though I feel this is important information to share. Does model performance and/or climate sensitivity differ between the two domains? Or do both domains yield very similar results (e.g., for T and LST)? And which domain approach do the authors recommend for future research in this region? Such information could be covered in the discussion section.

Response: This is a good question and was a concern of one of our co-authors. We carefully discussed among co-authors (and another senior researcher at MIT) on the concern of whether or not we should combine these 3 GLARM-large domain model results and 3 GLARM-small domain model results. We have agreed that a simple ensemble average seems questionable because these results are from two sampling groups that can possess different uncertainty distributions. We decided just to use one of the domains for simplicity. We

selected the small domain GLARM, which is similar to other RCM configurations for the Great Lakes climate studies, to represent the uncertainty inherited from different GCMs and enhance the computational efficiency. Nonetheless, please note that the results (GLARM-EA3) are similar to our previous 6-member ensemble results (GLARM-EA6), and all conclusions remain unchanged. Still, we did update the results (including numbers, figures, and tables) throughout the manuscript.

Specific comments

1. L1: No need to reply to this comment, but I am a little surprised that the authors choose GMD as a journal to publish their work. Given that the focus of the study is on the results of the future projections, I believe that a content-journals like for example ESD could have been a better fit for this work. That said, I respect the author's choice of GMD and do not suggest transferring this manuscript to a different journal.

2. L51: does this statement refer to an area or temporal change? Please clarify.

Response: this is corrected as "The overall ice coverage on the five Great Lakes has reduced by 71% from 1973 through 2010" (Wang et al. 2012).

3. L79-82: In this context, it could be interesting to check what the recent scientific results obtained as part of the ISIMIP lake sector tell for the Laurentian Great Lakes. See https://www.isimip.org/outcomes/publications-overview-page/

Response: in ISIMIP lake sector, we checked five relevant papers (listed below) (we already cited Woolway and Merchant 2019.) and we believe the most relevant one is the one we have cited (i.e. Woolway & Merchant, Worldwide alteration of lake mixing regimes in response to climate change, Nature Geoscience (2019) and second-most relevant one is Woolway et al., 2021 Phenological shifts in lake stratification under climate change Nature Communications, 12, 2318 (2021). It has been added to the reference in the revision.

The relevant findings in Woolway and Merchant 2019: By forcing the Flake model with four GCMs under two RCPs, Woolway and Merchant modeled and predicted the changes in the mixing regime of 635 lake around the world. Many lakes around the world are predicted to experience a reduction in mixing events such as the transition of monomictic lake to permanently stratified lakes and dimictic lakes to monomictic lakes. These future changes are expected to be driven by increase in lake surface temperature and significant decrease in winter ice cover duration.
In Woolway et al. (2021) they predicted the changes in the mixing regime of Northern Hemisphere lakes by forcing a four independently developed lake models with four GCMs,

each under three different RCP scenarios. They predict a longer thermally stratified season duration with earlier onset and later break-up, particularly for lakes situated at higher latitudes. The largest change in stratification phenology are projected under RCP 8.5 with stratification onset and break-up, respectively, occurring 22.0 ± 7.0 days earlier and 11.3 ± 4.7 days later, on average across the Northern Hemisphere.

Other papers:

Guo M., et al., Validation and Sensitivity Analysis of a 1-D Lake Model across Global Lakes, Journal of Geophysical Research: Atmospheres, 126, e2020JD033417 (2021)

Luke Grant et al. Attribution of global lake systems change to anthropogenic forcing
Nature Geoscience 14, pages 849–854 (2021) (2021)

Iestyn Woolway, et al.  Lake heatwaves under climate change.
Nature 589, 402–407 (2021)

4. L109: Importantly, RCP8.5 is not to be considered 'business as usual', but a 'high-end emission scenario'. And I suggest referring to RCP4.5 as a 'moderate mitigation scenario'

Response: Corrected as suggested.

5. L160-162: please add one or more refs to back this statement.

Response: Added : 1) Giorgi, F.: Thirty years of regional climate modeling: where are we and where are we going next?, Journal of Geophysical Research: Atmospheres, 124, 5696–5723, 2019.  2) Feser, F., Rockel, B., von Storch, H., Winterfeldt, J., Zahn, M., Feser, F., Rockel, B., Storch, H. v., Winterfeldt, J., and Zahn, M.: Regional climate models add value to global model data: a review and selected examples, B. Am. Meteorol. Soc., 92, 1181–1192, 2011.

6. Table 2: I suggest marking the selected GCMs in bold in this table

Response: Good suggestion. The selected GCMS are highlighted in bold.

7. Figure 3: caption and figure labels say 'GCM' but manuscript says 'AOGCM'. Please make this consistent (I think GCM is used more often nowadays).

Response: GCM is now used throughout the manuscript. AOGCM is removed.

8. Figure 4: Are the wintertime LSTs water temperatures taken only during the ice-free period or the average of the whole season (i.e., combined snow/ice/open water)? Please clarify. Also, caption says 'GLICD' but title of panel e says GLSEA. Also, has the acronym AICE (title panels e1-2) been introduced?

Response: the winter LSTs are the average for the whole season (combined snow/ice/open water), this is added in the caption. The panel legend is corrected as GLICD. AICE is removed and replaced with "ice cover" (AICE is the variable name in the model for Ice cover).

9. Figure 5: legend: GLARM(6MA), while text and other figures use GLARM-EA6. Also, spell out lake names (acronyms are not introduced in paper and they add no value, see also figure 12 and elsewhere)

Response: Lake names have been explicitly added to all applicable figures. And the typo of 6MA is corrected.

10. Figure 7 & 10: is this the standard deviation of the change (future minus past) or of the future state? Please clarify in the manuscript and/or caption

Response: they are the standard deviation of the change. (now the legend is revised as Std of T2 Change and Std of Precip Change), the caption is also updated as: The standard deviation of the projected ensemble changes in the annual mean surface air temperature over the Great Lakes basin during the mid-century (2030-2049) and the late century (2080-2099) in RCP 4.5 and RCP 8.5 scenarios, relative to the present-day climate (2000-2019)."

11. L327-328 & L332-333: I wonder if these changes (4 and 6%) are sufficiently different to say that RCP4.5 gives a stronger wetting than RCP8.5 for mid-century. Probably the uncertainty bands are largely overlapping? In that case I would rather say that they project a similar wetting.

Response: Agree. It is revised as "The projected mid-century increase in precipitation is similar in RCP 4.5 (6.5%) and RCP 8.5 (5.6%) with relatively similar atmospheric GHG concentrations over the period."

12. Fig. 14: to better understand the change, it would be more useful to also plot the present-day ice cover, or to plot the change in ice cover (future – present)

Response: Agree, we have added the present-day ice cover.

Textual comments

1. L26, 'are' > 'is'.

Response: corrected.

2. L61: 'will' > 'could'.

Response: corrected.

3. L72: 'predicted' > 'projected' (always use projections in the context of future climate) and 'atmospheric greenhouse gasses (GHGs)' > 'greenhouse gas (GHG) emissions'.

Response: corrected as suggested.

4. L89 and elsewhere: I'd suggest specifying 'Great Lakes' to 'Laurentian Great Lakes' throughout the manuscript, to avoid confusion with the African Great Lakes.

Response: The Laurentian Great Lakes are used at the beginning of abstract, Introduction and conclusion .

5. L101: FVCOM, this acronym hasn't been introduced yet

Response: Finite Volume Community Ocean Model (FVCOM) is defined in the revision.

6. L106: 'a RCM' > 'an RCM'

Response: corrected.

7. L116: 'LakesAtmosphere' > 'Lakes Atmosphere'

Response: corrected.

8. L116: 'iceatmosphere' > 'ice-atmosphere'

Response: corrected.

9. L155: 'projections' > 'assessment reports'

Response: corrected as suggested.

10. L233: 'predictions' > 'projections' (check elsewhere, predicted>projected) .

Response: corrected. We use projections in the context of future climate throughout the manuscript in the revision.

11. L252: remove space before '.' Check elsewhere for double or missing spaces.

Response: corrected.

12. L306: 'much more substantial' > 'more pronounced'

Response: corrected as suggested.

13. L309: 'cooling' > 'buffering'

Response: corrected as suggested.

14. L310: 'overlake' > 'over-lake'

Response: corrected as suggested.

15. L429: check punctuation

Response: corrected as suggested.

16. L428: check sentence (drop 'changes'?)
Response: This sentence is removed in the revision.

We hope we have addressed your questions satisfactorily. Thank you again for your time and efforts in reviewing our manuscript!

---

## Author Comment (AC6)

**Reviewer1:**

RC1: 'Comment on gmd-2021-440', Anonymous Referee #1, 04 Feb 2022
The paper presents climate projections of the Great Lakes region based on a regional climate model (GLARM) coupled with an ocean model (FVCOM) applied to the Great lakes. Climate projections derived from 3 ESMs and for two RCP scenarios have been used. The predictions for the mid and late 21st century are discussed. The results show the increased lake surface temperature and reduced ice cover at annual and seasonal scales with strongest changes over the Lake Superior.

Overall comments:
The paper is interesting and well written. The results are reasonable and well discussed. My main concern is the use of an ocean model to simulate the lake processes and the fact that the processes which are represented are not really described. It seems that only the energy transfers are represented and that there are no coupling with the surface model hydrology. I would like to know how the water volume of the lake is constrained, are they some glaciers melting water and lateral runoff inputs, water table exchanges ? how these contributions are impacted by climate warming? And how can they modify lake temperatures in addition to the direct exchanges with the atmosphere?

Response: Thanks for your question. The use of community ocean models to simulate the Great Lakes has been widely applied in an appropriate way.  Because of their sealike characteristics (including distant horizons, great depths, steep bathymetric gradients, strong Coriolis-influenced currents, and large thermal variability), the Great Lakes have long been referred to as ''inland seas'. All-natural water bodies (lakes and oceans) are physically described by the same set of primitive equations that are used in nearly all community ocean models. The major difference is that the Great Lakes is a freshwater system (no salinity simulation is needed, which is a standard option to turn on and off in all ocean models) and that's why it can be well handled by ocean models.

In fact, the NOAA official Great Lakes Operational Forecast System (GLOFS:https://tidesandcurrents.noaa.gov/ofs/glofs.html) utilizes FVCOM (Finite Volume Community Ocean Model) for the Great Lake hydrodynamic operational forecast. Note that FVCOM is also the hydrodynamic model we used in GLARM.  The use of ocean models (more precisely, they are all 3-D hydrodynamic models originally developed for ocean application but are also suitable for the Great Lakes ) in the Great Lakes has been for decades. Popular ocean models used for the Great Lakes include POM (Princeton Ocean Model), FVCOM (Finite Volume Community Ocean Model),  NEMO (Nucleus for European Modelling of the Ocean), etc. However, these models were applied to the Great Lakes in a standalone fashion. The importance of this study is the two-way coupling of the RCM (RegCM4) and 3-D hydrodynamic model (FVCOM) to resolve lake-air interactions to better represent the system for climate change projection, To date, no studies exist applying a 3-D

hydrodynamic model (only 1-D column lake models were used) two-way coupled with RCM to resolve the air-lake-ice interactions in projecting the evolution of the Great Lakes themselves interacting with regional climate changes.

Because there are plenty of applications and publications on the standalone hydrodynamic model (ocean model) simulations of the Great Lakes. We feel redundant to re-introduce and discuss them. FVCOM has gained popularity in research and applications to the Great Lakes (Anderson and Schwab, 2013; Bai et al., 2013; Beardsley et al., 2013; Xue et al. 2015; Anderson et al. 2018; Ye et al. 2020 and many more). There are other coastal hydrodynamic models (e.g., Beletsky et al., 2006; Fujisaki et al., 2013; Dupont et al., 2012; White et al., 2012 and others.) with similar characteristics to FVCOM, but we chose the FVCOM model because it is currently being used by NOAA for operational forecasting in the Great Lakes.

You are right, the hydrological cycle is not simulated in this paper for two reasons. First, surface hydrology requires great effort and needs to be studied separately. The water levels of the Great Lakes are primarily governed by the net basin supplies (NBS) of each lake (which are the sum of over-lake precipitation and basin runoff, and minus lake evaporation), in a combination with the Great Lakes regulation plan as well as inter-lake flows to describe a complete water budget. This requires a suite of models to be properly integrated to project water level changes. In fact, we have done it in our recent study of the Great Lakes water level, which is submitted to *Journal of Hydrology*, "Future Rise of the Great Lakes Water Levels under Climate Change" by Miraj B. Kayastha, Xinyu Ye, Chenfu Huang, Pengfei Xue* (corresponding author) (in revision). In which, we integrated GALRM (for over lake precipitation, evaporation), LBRM(Large Basin Runoff Model for river runoff into each lake ), CGLRRM (Coordinated Great Lakes Regulation and Routing Model for inter-lake flow and regulation plans) to project the changes in surface hydrology and the Great Lakes water level change in the future. Given the complexity and importance of this topic, it is beyond the scope of this study. Second, the water level fluctuation (1-2 m) does not impact the surface area of the Great Lakes (considering the depth and size of these lakes), therefore, water level change (which is critical for coastal erosion, navigation) does not play an important role in influencing lake-air heat fluxes and climate change, that's why we simulate the over lake evaporation (latent heat flux) but did not simulate complete surface hydrological cycle in this study.

These are now explicitly mentioned in the discussion section "*We note that this study does not directly simulate the surface hydrological cycle for three reasons. First, the water levels of the Great Lakes are primarily governed by the net basin supply (NBS) of each lake (over-lake precipitation, river runoff, and lake evaporation), in combination with natural and regulated inter-lake flows. The projection of water level changes requires the*

*integration of a suite of models. Such integration is documented in our separate study (Kayastha et al., under review), in which we use GLARM (for over-lake precipitation, lake evaporation), LBRM (Large Basin Runoff Model) for river runoffs into each lake, CGLRRM (Coordinated Great Lakes Regulation and Routing Mode) for inter-lake flows. Given the complexity of the projection of the surface hydrological cycle, it is beyond the scope of this study. Second, the impact of water level change on the surface area of the Great Lakes is negligible; therefore, water level change does not play a critical role in influencing lake-air heat fluxes and climate change. Third, compared to the primary factor (surface heat fluxes) of lake thermal change, the heat transport between lakes associated with inter-lake flows is secondary on the lake basin-wide scale. It falls in the uncertainty of surface heat fluxes in the GLARM projections.*"

Anderson, E. J., Fujisaki-Manome, A., Kessler, J., Lang, G. A., Chu, P. Y., Kelley, J. G., ... & Wang, J. (2018). Ice forecasting in the next-generation great lakes operational forecast system (GLOFS). *Journal of Marine Science and Engineering*, *6*(4), 123.

Anderson, E. J., & Schwab, D. J. (2013). Predicting the oscillating bi-directional exchange flow in the Straits of Mackinac. *Journal of Great Lakes Research*, *39*(4), 663-671.

Bai, X., Wang, J., Schwab, D. J., Yang, Y., Luo, L., Leshkevich, G. A., & Liu, S. (2013). Modeling 1993–2008 climatology of seasonal general circulation and thermal structure in the Great Lakes using FVCOM. *Ocean Modelling*, *65*, 40-63.

Beardsley, R. C., Chen, C., & Xu, Q. (2013). Coastal flooding in Scituate (MA): A FVCOM study of the 27 December 2010 nor'easter. *Journal of Geophysical Research: Oceans*, *118*(11), 6030-6045.

Beletsky, D., Schwab, D., & McCormick, M. (2006). Modeling the 1998–2003 summer circulation and thermal structure in Lake Michigan. *Journal of Geophysical Research: Oceans*, *111*(C10).

Dupont, F., Chittibabu, P., Fortin, V., Rao, Y. R., & Lu, Y. (2012). Assessment of a NEMO-based hydrodynamic modelling system for the Great Lakes. *Water Quality Research Journal of Canada*, *47*(3-4), 198-214.

Fujisaki, A., Wang, J., Bai, X., Leshkevich, G., & Lofgren, B. (2013). Model-simulated interannual variability of Lake Erie ice cover, circulation, and thermal structure in response to atmospheric forcing, 2003–2012. *Journal of Geophysical Research: Oceans*, *118*(9), 4286-4304.

White, B., Austin, J., & Matsumoto, K. (2012). A three-dimensional model of Lake Superior with ice and biogeochemistry. *Journal of Great Lakes Research*, *38*(1), 61-71.

Xue, P., Schwab, D. J., & Hu, S. (2015). An investigation of the thermal response to meteorological forcing in a hydrodynamic model of L ake S uperior. *Journal of Geophysical Research: Oceans*, *120*(7), 5233-5253.

Ye, X., Chu, P. Y., Anderson, E. J., Huang, C., Lang, G. A., & Xue, P. (2020). Improved thermal structure simulation and optimized sampling strategy for Lake Erie using a data assimilative model. *Journal of Great Lakes Research*, *46*(1), 144-158.

Specific comments:

- Equation 1, P7: I am not familiar with these scores, is there a justification to have this form of combination of the metrics? I am wondering if the exponent should be 1/(m+n) instead of 1/(mxn)? Is there a reference to this equation that could be added?

Response: It is 1/(mxn), As we cited in our paper, this method including the equation is documented in (Giorgi and Mearns, 2002, Journal of Climate: Calculation of Average, Uncertainty Range, and Reliability of Regional Climate Changes from AOGCM Simulations via the "Reliability Ensemble Averaging" (REA) Method). This is equation (4) in their paper.

The equation is also shown as equation 2 in Miao, C., Duan, Q., Sun, Q., Huang, Y., Kong, D., Yang, T., Ye, A., Di, Z., and Gong, W.: Assessment of CMIP5 climate models and projected temperature changes over Northern Eurasia, Environmental Research Letters, 9, 055 007, 2014. (this citation is also added in the revision)

-P12, Table 2 , REA is not defined, how did you combined the 3 statistical metrics ?

Response: REA is defined in line 188 in the original manuscript, but we noticed that the description was not clear. We have revised the description of GCM selection (i.e. the description of equations 1 and 2 for reliability analysis) in line 135-145 in the revised manuscript, it should be clear. Regarding table 2, we shouldn't put "REA" there, it should be "normalized reliability score" (which has caused your confusion), this has been corrected in the revision.

Notice that the reliability analysis was used to select the three GCM models. AFTER the three GCMs are selected, we then used taylor diagrams (RMSD, correlation, Std; figure 2 upper panel) and warming trend analysis (figure 2, lower panel ) to check (validate) if our GCM selection is appropriate. The reliability analysis for GCM selection does not (should not) combine the 3 statistical metrics. The three statistic metrics are for independent validation of our GCM selection.

-P14, line 263, evaporation and latent heat flux are the same variable (in different units) please modify your sentence.

Response: Within the Great Lakes, LST and ice cover are the two most important physical lake variables that influence the lake-atmosphere heat and water fluxes by affecting solar radiation, longwave radiation, and sensible and **latent (evaporation) heat**.

- Figure 4: it would be more clear to map the differences mod/obs in the right column

Response: We (co-authors) have discussed this comment internally. We feel that, for the present-day simulation, we prefer to show the model simulated patterns to give readers an

intuitive feeling of the model performance. In addition, we discussed modeled patterns (not only differences) in the later sections, so showing the observed and model pattern rather than the differences is better.  We prefer to retain the current plot. Thanks for your understanding.

-Figure 5: the names of the lakes need to be added on the plots,

Response: added.

-Figure 6: the legend is not clear, is it annual mean of the differences that are plotted, what about seasonal variations?

Response: Thanks, it is the annual mean of the differences. We changed the caption into "*The projected changes in the annual mean surface air temperature over the Great Lakes basin during the mid-century (2030-2049) and the late century (2080-2099) in RCP 4.5 and RCP 8.5 scenarios, relative to the present-day climate (2000-2019).*" And the legend has also been revised as "T2 change" instead of "T2", this should avoid any potential confusion.  The seasonal variations are presented and discussed in  Figure 7 and Table 3. Notice that we have had results for  2 scenarios and 2 periods and we already have 16 figures (we added more important information on lake thermal structure) and 5 tables. We also try to pick the most important information for readers and avoid overwhelming them.

-Figure9: Total precipitation changes are plotted, how is it shared between rainfall and snowfall? How rainfall is treated when it falls over the lake? Can it freeze/melt when the lake is ice covered?

Response:  This study doesn't distinguish between rainfall and snowfall and doesn't include its impact on water level change (please see our response to the general comment).

- Figure 13 and text related: do you have explanations concerning the lower warming of the Erie lake? The lake is the shallowest, it should be more impacted by the atmosphere warming, did I miss something?

Response: Good question!  In the revised version, one of the major changes we made is to address this question. We have dedicated a thorough discussion from lines 279-302 and new figure 12 (projected thermal structure change) to this. Here is  a short summary: It is related to the strong early stratification in the deep lakes that cause a significant increase in spring LST. And the higher ice cover in Lake Erie (which leads to a relatively lower increase in LST during winter), and relatively lower ice in deep lakes. This is because deep lakes are, by nature, large heat reservoirs that can transfer heat from a deep lake layer to the surface to reduce ice formation. The best example is the observed ice coverage of the shallowest lake (Erie) and the second deepest lake (Ontario). Both lakes have small surface areas but significantly different water depths (mean water depths are 19 m and 86 m, respectively, Fig. 1, panel b), resulting in high (low) winter ice cover in Lake Erie (Ontario) (Fig. 4).

Citation: https://doi.org/10.5194/gmd-2021-440-RC1

A note:

Finally, we want to let you know that in response to other reviewer's question and our (Co-PI) internal discussion (also in consulting with a senior climate scientist at MIT) on the concern of whether or not we should combine these 3 GLARM-large domain model results and 3 GLARM-small domain model results. We agreed that a simple ensemble average seems questionable because these results are from two sampling groups that can possess different uncertainty distributions.  We decided just to use one of the domains. We selected the small domain GLARM, which is similar to other RCM configurations for the Great Lakes climate studies,  to represent the uncertainty inherited from different GCMs and enhance the computational efficiency. Nonetheless, please note that the results (GLARM-EA3) are similar to our previous 6-member ensemble results (GLARM-EA6), and all conclusions remain unchanged. We have updated the results (like numbers, figures, and tables) throughout the manuscript, please see track change version that marks  all updates and changes.

Thank you again for your questions and suggestions. I hope we have addressed your concerns and questions satisfactorily.